# Haploinsufficiency of the lysosomal sialidase *NEU1* results in a model of pleomorphic rhabdomyosarcoma in mice

Eda R. Machado[1,7], Diantha van de Vlekkert [1,7], Heather S. Sheppard [2,7], Scott Perry[3,7], Susanna M. Downing[4], Jonathan Laxton[3], Richard Ashmun[3], David B. Finkelstein[5], Geoffrey A. Neale [6], Huimin Hu[1], Frank C. Harwood[1], Selene C. Koo [2], Gerard C. Grosveld [1✉] & Alessandra d'Azzo [1✉]

Rhabdomyosarcoma, the most common pediatric sarcoma, has no effective treatment for the pleomorphic subtype. Still, what triggers transformation into this aggressive phenotype remains poorly understood. Here we used $Ptch1^{+/−}/ETV7^{TG/+/−}$ mice with enhanced incidence of rhabdomyosarcoma to generate a model of pleomorphic rhabdomyosarcoma driven by haploinsufficiency of the lysosomal sialidase neuraminidase 1. These tumors share mostly features of embryonal and some of alveolar rhabdomyosarcoma. Mechanistically, we show that the transforming pathway is increased lysosomal exocytosis downstream of reduced neuraminidase 1, exemplified by the redistribution of the lysosomal associated membrane protein 1 at the plasma membrane of tumor and stromal cells. Here we exploit this unique feature for single cell analysis and define heterogeneous populations of exocytic, only partially differentiated cells that force tumors to pleomorphism and promote a fibrotic microenvironment. These data together with the identification of an adipogenic signature shared by human rhabdomyosarcoma, and likely fueling the tumor's metabolism, make this model of pleomorphic rhabdomyosarcoma ideal for diagnostic and therapeutic studies.

[1] Department of Genetics, St. Jude Children's Research Hospital, Memphis, TN 38105, USA. [2] Department of Pathology, St. Jude Children's Research Hospital, Memphis, TN 38105, USA. [3] Flow Cytometry Core Facility, St. Jude Children's Research Hospital, Memphis, TN 38105, USA. [4] Department of Cell & Molecular Biology, St. Jude Children's Research Hospital, Memphis, TN 38105, USA. [5] Department of Computational Biology, St. Jude Children's Research Hospital, Memphis, TN 38105, USA. [6] Hartwell Center for Bioinformatics and Biotechnology, St. Jude Children's Research Hospital, Memphis, TN 38105, USA. [7] These authors contributed equally: Eda R. Machado, Diantha van de Vlekkert, Heather S. Sheppard, Scott Perry. ✉email: gerard.grosveld@stjude.org; sandra.dazzo@stjude.org

Genetic, epigenetic, and environmental cues shape molecularly distinct subpopulations of cells in an evolving solid tumor[1,2]. This intratumoral heterogeneity is a likely prelude to chemotherapy resistance and metastases. Rhabdomyosarcoma (RMS) is the most common pediatric soft tissue sarcoma with poor outcomes for high-risk patients[3]. Although it usually displays myogenic features, RMS can originate from both myogenic and non-myogenic progenitor cells, which explains, at least in part, its different sites of occurrence (e.g., limbs, chest, head and neck, and retroperitoneum)[4]. Generally, these tumors are classified into two major histological subtypes: (1) ERMS (embryonal RMS) (~60%) has a mostly favorable outcome, which decreases to only 40% overall survival when metastatic; (2) ARMS (alveolar RMS) (~20%) is the aggressive form associated with early metastatic dissemination and poor prognosis[4]. More than 80% of ARMS are defined by two chromosome translocations, involving *PAX3/7* and *FOXO1*[4,5]. Both ERMS and the rare pleomorphic RMS subtypes lack these chromosome translocations and develop into genetically more complex tumors, frequently showing inactivation of the p53 pathway and/or another oncogenic driver mutations[6–12]. These tumors express both developmentally early myogenic markers (e.g., PAX3/7, MyoD, and Myogenin) and markers of terminal differentiation (e.g., Desmin and SMA)[13–15]. Although several mouse models of ERMS have been reported[16–18], none addressed the molecular and cellular events that drive RMS into a pleomorphic state, which therefore remains poorly understood.

We have shown earlier that haploinsufficiency of the gene encoding the lysosomal sialidase *Neu1* in *Arf*[−/−] mice rendered different types of sarcomas more aggressive[19]. NEU1 hydrolyzes terminal sialic acids from sialylated glycoproteins, changing their biochemical properties and affecting downstream pathways[20,21]. One of these pathways is lysosomal exocytosis which NEU1 negatively regulates by cleaving the sialic acids of LAMP1. This lysosomal membrane protein is responsible for docking lysosomes at the plasma membrane (PM) of cells, prior to their fusion with the PM[22–24]. Low NEU1 activity increases the number of lysosomes, decorated with a long-lived, sialylated LAMP1, that dock at the PM. This ultimately results in excessive lysosomal exocytosis with deleterious consequences for the integrity of PMs and the extracellular matrix (ECM)[19,22,24]. The common readouts of this aberrant process in cells are increased activity of lysosomal enzymes (e.g., β-hexosaminidase) extracellularly and relocation of LAMP1 to the PM[22,25].

Dysregulated lysosomal exocytosis is the basis of disease pathogenesis in *Neu1*[−/−] mice, a model of the lysosomal storage disease sialidosis[20,22,26,27]. Relevant to these studies is the muscle connective tissue phenotype in *Neu1*[−/−] mice that undergo a precancerous transformation, leading to expansion of the tissue via increased proliferation and generalized fibrosis[26,27]. *Neu1* deficient fibroblasts behave as myofibroblasts/mesenchymal cells, being simultaneously proliferative and migratory/invasive. They show increased exocytosis of soluble proteolytic enzymes and exosomes, which together remodel the ECM and propagate the fibrotic disease[26]. We found that this pathogenic cascade accelerated the occurrence of several types of sarcomas in *Neu1*[+/−]/*Arf*[−/−] mice and transformed these tumors into an aggressive and chemoresistant phenotype[19]. We sought to explore these Neu1-mediated phenomena in a mouse model of RMS.

The *Patched1* (*Ptch1*)[+/−] mouse, a model of medulloblastoma, also develops ERMS-like tumors at low incidence[9,28,29]. These tumors are driven by activation of the Shh pathway[9,28,30], which is normally kept inactive by binding of the Ptch1 receptor to Smo (smoothened)[29,31]. Reduced Ptch1 expression frees Smo, which then hyperactivates Gli transcription factors, triggering growth and proliferation. Crossing the *Ptch1*[+/−] mice with a transgenic

mouse line (*ETV7*[TG+/−]) expressing human ETV7 increased the incidence and penetrance of ERMS and hematopoietic malignancy[32,33]. ETV7, an ETS transcription factor[34], promotes the assembly of a novel rapamycin-resistant mTOR complex, mTORC3, by directly binding to mTOR[32]. ETV7 is differentially expressed in all human RMS, and mTORC3 increases the proliferation of several human ERMS cell lines. Furthermore, it was shown that the higher incidence of RMS in *Ptch1*[+/−]/*ETV7*[TG+/−] mice is driven by mTORC3[32].

Here, we show that reducing Neu1 expression in the *Ptch1*[+/−]/*ETV7*[TG+/−] model changes RMS tumors into a pleomorphic state. These tumors have increased cell heterogeneity, a distinctly fibrotic microenvironment, and share features with both human ERMS and ARMS. We found that deregulated lysosomal exocytosis downstream of low Neu1 is the underlying pathway driving transformation in these tumors, which are also fueled by an adipogenic signature. Thus, *Neu1*[+/−]/*Ptch1*[+/−]/*ETV7*[TG+/−] mice are a model of human pleomorphic RMS and may represent a powerful means for diagnostic and therapeutic studies of these aggressive and incurable tumors.

## Results

**Neu1 haploinsufficiency promotes the development of poorly differentiated RMS in the *Ptch1*[+/−]/*ETV7*[TG+/−] mouse model.** In *Neu1*[+/−]/*Ptch1*[+/−]/*ETV7*[TG+/−] (NPE) mice, low *Neu1* RNA and protein expression (Supplementary Fig. 1a, b) was sufficient to increase the incidence of RMS to 62%, as compared to 54% in the *Ptch1*[+/−]/*ETV7*[TG+/−] (PE) mice and only 8% in the *Ptch1*[+/−] mice in 2 years (Fig. 1a). RMS was diagnosed based on morphology and immune reactivity to the myogenic regulatory factors (Mrfs) MyoD and Myogenin, and the muscle-specific type III intermediate filament desmin (Des) (Supplementary Fig. 1c–e)[35]. The survival rate of both NPE and PE mice was comparable but significantly lower than that of *Ptch1*[+/−] mice (Fig. 1b). All RMS tumors in NPE and PE models developed in either the extremities or the trunk, reflecting the location of human ARMS and pleomorphic RMS with poor prognosis and short survival[4] (Fig. 1c). Within the cohort of mice with RMS tumors 4/35 (11.4%) NPE and 1/23 (4.3%) PE mice also developed secondary growths (Fig. 1a, c), a finding that may indicate metastatic spread, although without additional genomic analysis we cannot exclude that these tumors are synchronous rather than metachronous. We next established that *ETV7* was expressed in both NPE and PE tumors, as compared to *Neu1*[+/−]/*Ptch1*[+/−] (NP) controls (Fig. 1d). We also confirmed that ETV7 expression in both models supported the assembly of mTORC3, as shown by two-way co-immunoprecipitation of ETV7 and mTOR (Fig. 1e, f and Supplementary Fig. 2a, b). In addition, the PP2Ac and phospho-4E-BP1[Thr37/46] components of mTORC3, also co-immunoprecipitated with mTOR and ETV7 (Fig. 1f and Supplementary Fig. 2b), further supporting the presence of the complex. These results suggest that the rapid growth of RMS in NPE and PE mice is driven in part by mTORC3 activity, although we cannot exclude the possible contribution of *Neu1* heterozygosity, giving these tumors an additional proliferative advantage.

What distinguished NPE from PE tumors was their histopathological appearance. RMS in the NPE model showed pronounced cellular heterogeneity, associated with the presence of numerous undifferentiated, anaplastic foci, rich in pleomorphic and rhabdoid cells, as compared with the more differentiated morphology of PE RMS (Fig. 1g, h, Supplementary Fig. 1c, d). These pleomorphic foci were populated with intertwining bundles of tumor cells of abnormal size and shape (Fig. 1h arrows), and small undifferentiated tumor cells (Fig. 1h arrowhead).

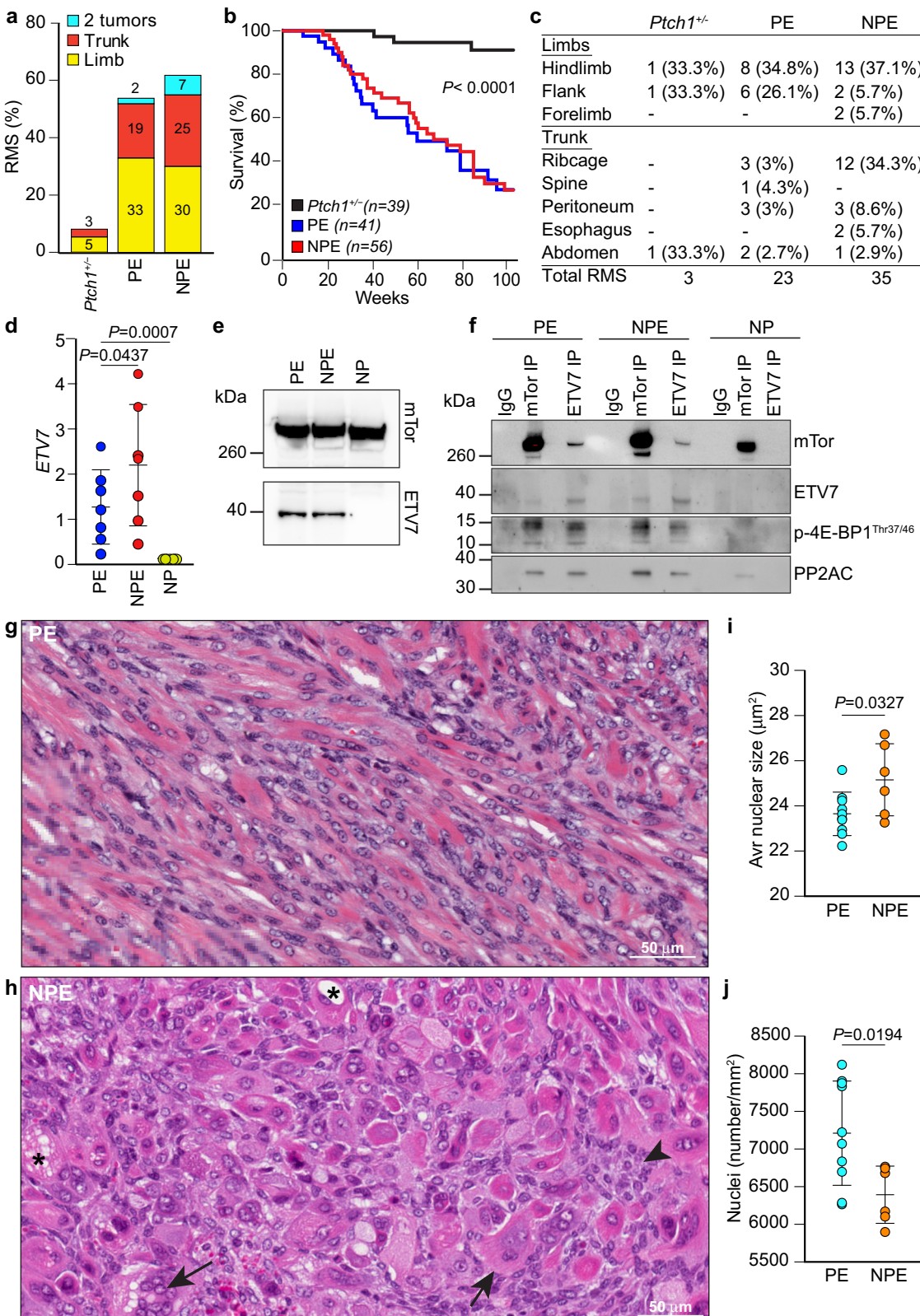

An algorithm designed to quantify nuclear morphometry in tumor H&E sections revealed an increased number of aberrantly large, multinucleated cells with atypical mitotic figures, predominantly in NPE tumors (Fig. 1i). Because of the very large size of most of the nuclei in the NPE tumors, their total number per annotated area was lower than that in PE tumors (Fig. 1j). Instead, other typical RMS features were common to both NPE

and PE tumors, including spindle-shaped cells, round rhabdo-myoblasts, cross-striated mature skeletal myocytes and myofibers (Fig. 1g, h), as well as expression of MyoD, Myogenin, and Desmin (Supplementary Fig. 1e)[35]. Together, these results suggest that *Neu1* haploinsufficiency superimposed on the spontaneously occurring PE tumors changes the overall morphology and differentiation status of RMS.

**Fig. 1 Undifferentiated RMS arises in NPE mice. a** Percentages of RMS types developed by *Ptch1*[+/−], PE, and NPE mice. **b** Percent survival of *Ptch1*[+/−], PE, and NPE mice. *P* calculated with log-rank test for trend. **c** Location, number and percentage of RMS developed in *Ptch1*[+/−], PE, and NPE mice. **d** *ETV7* mRNA expression in PE and NPE tumors. NP: *Neu1*[+/−]/*Ptch1*[+/−] tumors were used as control. Mean ± s.d.; One-way ANOVA *t*-test; *n* = 7 biologically independent tumors. **e** Western blot of mTOR and ETV7 in PE, NPE, and NP tumors. **f** Two-way co-immunoprecipitation of mTOR and ETV7 and co-immunoprecipitation of mTOR and ETV7 with mTORC3 components, PP2Ac and phospho-4E-BP1[Thr37/46], in PE and NPE tumors. NP tumors were used as control. **g, h** Morphological characteristics of PE (**g**) and NPE (**h**) tumors by H&E staining. Arrow—large abnormal and multinucleated cancer cells; Arrowhead—undifferentiated small cancer cells; asterisk—cytoplasmic vacuolation consistent with accumulation of neutral lipids. Scale bar: 50 μm. **i, j** Quantification of the size of nuclei (**i**) and quantification of number of nuclei (**j**) in tumor sections of NPE and PE mice. Mean ± s.d.; Student's (unpaired) *t*-test; PE: *n* = 10 and NPE: *n* = 6 biologically independent tumors.

**Neu1-mediated pleomorphism is associated with neoplastic fibrosis.** In general, aggressive/invasive tumors undergo active remodeling of their ECM, which creates a fibrotic micro-environment that facilitates metastatic spread and promotes drug resistance[36]. To assess the extent of collagen deposition and ECM remodeling, Masson's trichrome staining identified extensive areas of collagenous connective tissue that were much more prominent in the NPE (blue) than in PE tumors, particularly at invasive fronts and pleomorphic foci (Fig. 2a, b). A similar increase in connective tissue was observed in Masson's trichrome-stained human tissue microarrays (TMAs) from several RMS patients (ERMS *n* = 16, ARMS *n* = 17, and spindle cell/sclerosing RMS *n* = 3), where 50% of all TMA cores had more than 9% of the total area as collagenous material, with an average of 28.7% in ERMS, 9.7% in ARMS, and 23.3% in spindle cell/sclerosing RMS, although the latter only comprised three samples (Fig. 2c and Supplementary Fig. 3a). In addition, we observed a substantial number of cells that showed intracytoplasmic collagen as well, with an average of 12.8% in ERMS, 8.5% in ARMS, and 13.7% in spindle cell/sclerosing RMS (Supplementary Fig. 3b). Quantification of the staining of the mouse tumors revealed a significant increase of connective tissue in the NPE tumors compared to PE tumors, underscoring their more aggressive nature (Fig. 2d). Similar to what is seen in human RMS tumors, subpopulations of NPE cells also showed quantifiable intracytoplasmic collagen (Fig. 2e and Supplementary Fig. 3b), indicating their cell-autonomous capacity to deposit ECM, like that of myofibroblasts during fibrosis[26,36]. The latter was further confirmed by the significant upregulation of *Col1a2* and *Col4a1* expression in NPE versus PE RMS (Fig. 2f, g). These combined observations were not seen in normal muscle tissue of these mice (Supplementary Fig. 3c–e), emphasizing that the connective tissue deposition observed in the NPE RMS tumors was caused by the combination of reduced activity of Neu1 within tumor cells and stroma. Thus, reduced Neu1 expression promotes pleomorphism and transforms the tumor stroma into a desmoplastic, fibrotic state.

***Neu1*[+/−]/*Ptch1*[+/−]/*ETV7*[TG+/−] mice share molecular characteristics of human ARMS and ERMS.** We next compared the gene expression profiles of NPE and PE tumors with RNA-seq data from human ERMS and ARMS available from the St Jude Pediatric Cancer Genome Project (PCGP). As shown in the heatmap (Fig. 2h), the pool of human genes that could be analyzed and compared were 91 genes from different platforms (murine microarray and Pediatric Cancer Genome Project (PCGP) RNAseq) that passed the principal component analysis (PCA), quality control, and significant Pearson correlation criteria. This comparative analysis showed no significant differences between human RMS and our NPE and PE tumors. Within this geneset, the highest positive correlation was found between the expression profiles of NPE (*r* = 0.71) and PE (*r* = 0.72) tumors with human ERMS that comprise the most genetically heterogeneous subtypes[4] (Fig. 2h). Although no significant differences at the transcriptional levels were observed between NPE or PE

tumors compared to human RMS, our data support the notion that *Neu1* haploinsufficiency in the NPE model promotes the development of a more poorly differentiated phenotype, as demonstrated by their histologically pleomorphic appearance. Enrichment analysis of NPE/PE and ERMS correlative genes identified pathways involved in metabolism, cell signaling, and differentiation (Supplementary Fig. 4a, b, Supplementary Data 2). This analysis also distinguished two small clusters of genes in NPE and PE tumors that correlated with the expression signatures of *PAX3-FOXO1* ARMS (*r* = 0.67 and *r* = 0.68, respectively) and *PAX7-FOXO1* ARMS (*r* = 0.65 and *r* = 0.66, respectively) (Fig. 2h, orange boxes). These clusters included *PAX7* and *CTHRC1* (collagen triple helix repeat containing 1), which encode proteins involved in cell differentiation and tissue remodeling. Together, this comparison highlights the genetic resemblance between the murine NPE and PE tumors with human RMS, in particular with the ERMS subtype.

**Low NEU1 expression correlates with plasma membrane redistribution of LAMP1 and increased lysosomal exocytosis.** To assess the molecular events that might contribute to the aberrant histopathology and fibrotic state of NPE tumors, we focused on exacerbated lysosomal exocytosis. Testable readouts of this dysregulated process are an accumulation of a long-lived, sialylated LAMP1 due to low NEU1 activity, and LAMP1 redistribution at the cell surface[22,25]. We exploited both these parameters to measure the expression levels of LAMP1 as well as the fraction of this protein at the cell surface in human TMAs from several RMS patients. Matched immunostained tissue cores from RMS patients showed that very low expression of NEU1 was accompanied by an overall increase in LAMP1 (Fig. 3a, Supplementary Fig. 5a, b). Using a membrane immunoreactivity computer learning algorithm applied to TMA cores, we quantified LAMP1 positive staining at the cell membrane (LAMP1[PMpos]) (Fig. 3b–d). We found that in most RMS samples, independent of their subtype, low NEU1 expression correlated with increased LAMP1 at the cell surface (Fig. 3d), a potential indicator of the cells' enhanced exocytic activity. Additionally, analysis of LAMP1 immunoreactivity in each TMA via a density heatmap identified areas of dense immunoreactivity for LAMP1 that mostly coincided with tumor-stromal borders and areas of pleomorphism (Fig. 3b, c). This patchy LAMP1 positive staining also highlighted the extent of cell heterogeneity within each RMS sample (Fig. 3c).

Increased LAMP1 levels in human RMS were paralleled by a similarly altered expression pattern of this protein in NPE compared with PE RMS (Fig. 3e). To assess how many mouse tumor cells had increased Lamp1[PMpos], as a proxy for enhanced lysosomal exocytosis, we performed flow cytometric analysis of non-permeabilized NPE and PE dissociated tumor cells (Fig. 3f, g). This identified a population of NPE tumor cells with high Lamp1[PMpos], which was distinct from that in PE tumors (Fig. 3f, g). To prove that NPE RMS had increased lysosomal exocytosis, interstitial fluids from both NPE and PE tumors were assayed for the lysosomal enzyme β-hexosaminidase[22,37]. Significantly higher

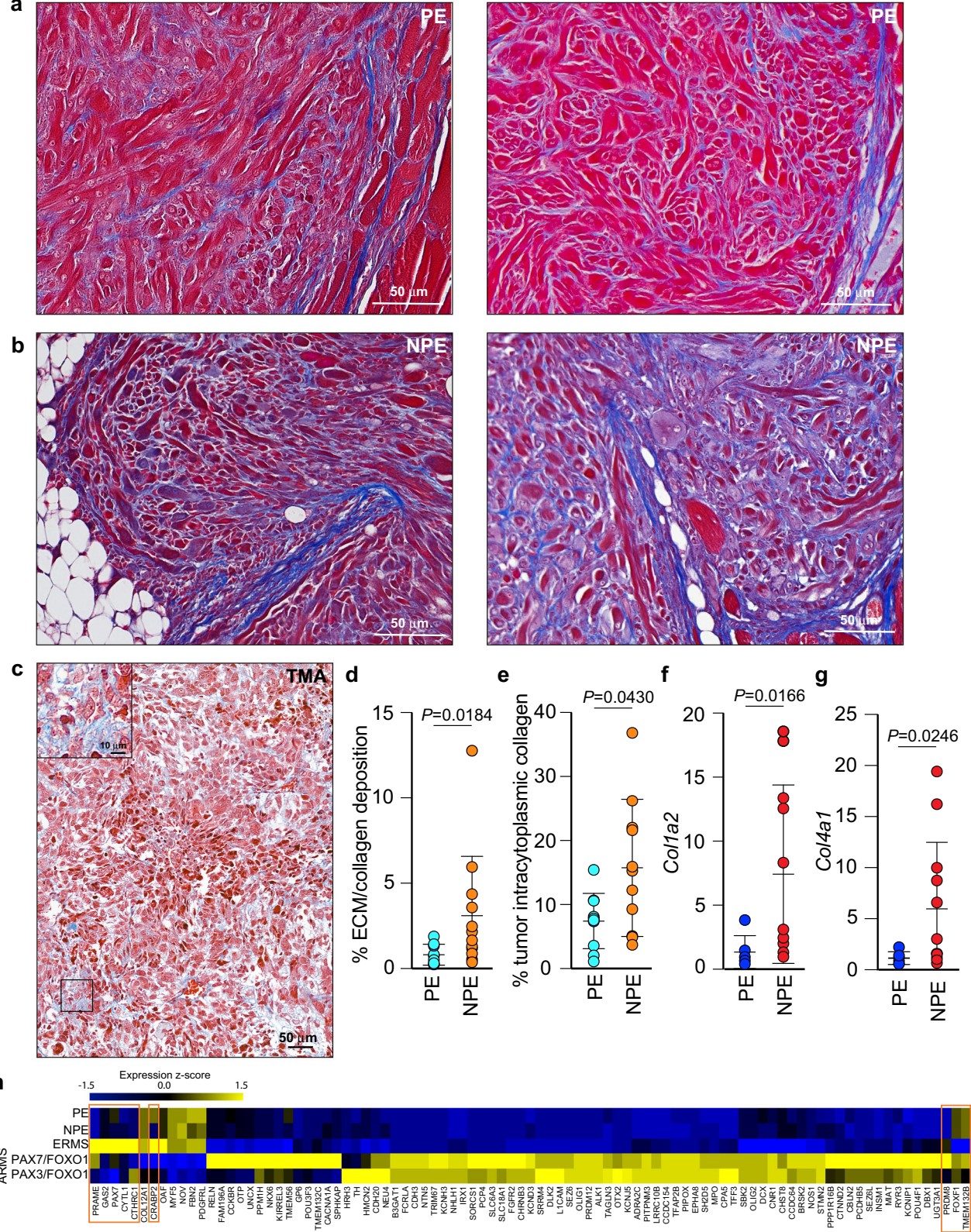

activity of β-hexosaminidase was measured in the interstitial fluid of several NPE RMS compared with PE samples (Fig. 3h), confirming that NPE tumors were more exocytic.

**Single-cell multiplex flow cytometry underscores the complex cell heterogeneity in $Neu1^{+/-}/Ptch1^{+/-}/ETV7^{TG+/-}$ tumors.** Based on the combined features of increased Lamp1$^{PMpos}$,

lysosomal exocytosis, and pleomorphism in limb and trunk NPE compared with PE RMS, we designed a multiplex high parameter flow cytometry strategy that evaluated Lamp1$^{PMpos}$ cells separately from Lamp1$^{PMneg}$ cells within three main cell populations: cancer, stroma, and hematopoietic cells (Fig. 4a). Protein expression data were generated by using Lamp1$^{PM}$ in combination with markers selective for each cell population[35,38–50],

**Fig. 2 Pleomorphism is associated with fibrosis and murine RMS resembles human RMS. a, b** Masson's Trichrome staining of PE RMS (**a**) and NPE (**b**) RMS depicting ECM/collagenous material in blue and cell cytoplasm in red. Scale bar: 50 μm. **c** Representative micrograph of one core of human RMS TMAs (n = 36 biologically independent samples) stained with Masson's Trichrome. Scale bar: 50 μm, inset scale bar: 10 μm. **d** Quantification of ECM/collagen deposition in PE and NPE RMS; Mean ± s.d.; Mann–Whitney (unpaired) t-test; PE: n = 9 and NPE: n = 12 biologically independent tumors. **e** Quantification of tumor intracytoplasmic collagen from PE and NPE RMS; Mean ± s.d.; Mann–Whitney (unpaired) test; PE: n = 10 and NPE: n = 11 biologically independent tumors. **f** PE and NPE *Col1a2* mRNA expression relative to PE. Mean ± s.d.; Welch (unpaired) t-test; PE: n = 6 and NPE: n = 11 biologically independent tumors. **g** PE and NPE *Col4a1* mRNA expression relative to PE. Mean ± s.d.; Welch (unpaired) t-test; PE: n = 6 and NPE: n = 12 biologically independent tumors. **h** Heatmap of gene expression comparison between murine and human RMS. The nine genes that correlated between murine RMS and ARMS are boxed in orange. Pearson's correlation was performed on 91 genes analyzed.

followed by tSNE analysis. The distribution and density of cells in limb and trunk RMS from NPE and PE mice highlighted the extent of cell heterogeneity (Fig. 4b). To uncover distinct cell populations within each tumor sample, we narrowed the tSNE analysis to tumor and stromal cells by excluding hematopoietic cells, using CD45 (Supplementary Fig. 6). Several discrete cell clusters were identified, which differed not only between NPE and PE RMS, independently of their CD45 status, but also between limb and trunk tumors (Fig. 4c). This may be linked to the cell of origin that supports the development of trunk versus limb tumors, but a more comprehensive investigation would be necessary to confirm these differences.

**Lamp1 at the plasma membrane marks highly exocytic and undifferentiated cell populations in Neu1⁺/⁻/Ptch1⁺/⁻/ETV7^TG+/⁻ tumors.** tSNE analyses of the selected CD45^neg cell populations in limb and trunk tumors of both NPE and PE mice (Fig. 4b, c) again revealed distinct cell density and distribution (Fig. 5a, b). These cell populations were further analyzed by using Lamp1^PM expression in combination with a set of canonical markers distinct for cancer and stromal cells (i.e., desmin, Ki67, Sca1, Sma, Pcam1, CD44, Lyve1, and EpCam) (Fig. 5c–h). This allowed for the annotation of 26 and 22 different cell clusters in limb and trunk tumors, respectively (numbered in Fig. 5c–h). Within the annotated populations, the total percentage of Lamp1^PMpos exocytic cells was 45.95% in limb (cell clusters 1–12) and 11.14% in trunk (cell clusters 1–10) RMS (Fig. 5e–h). In both limb and trunk RMS a clearly larger percentage of Lamp1^PMpos cells was detected in NPE (cluster 1–12) than in PE (cell clusters 1–10) tumors (Fig. 5e, f). Most importantly, the majority of Lamp1^PMpos cells showed pleomorphic features (cell clusters 1–6) and expressed simultaneously the terminal (myo)differentiation marker desmin and markers of progenitor or undifferentiated cells, such as Ki67 (proliferation), Sca1 (progenitor), Sma (smooth muscle actin/mesenchymal), Pcam1 and CD44 (endothelial), and Lyve1 (lymphatic endothelial) (Fig. 5c–f). These exocytic, pleomorphic subpopulations accounted for 21% and 15% of cancer cells in the limb and trunk (cell clusters 1–6), respectively, and were more abundant in NPE than PE RMS (Fig. 5e–h). The combination of pleomorphic and exocytic cells enriched in NPE tumors indicates that *Neu1* haploinsufficiency promotes the poorly differentiated state of the tumors, albeit an in-depth explanation of this phenomenon may require further investigation.

Similarly, within the stromal CD45^neg and Des^neg cell populations, the majority of Lamp1^PMpos cells included cancer-associated fibroblasts (SMA) (cell cluster 7–9) and epithelial cells (cell clusters 10–11 in limb and 10 in trunk), but also expressed markers typical of myofibroblasts/mesenchymal cells, e.g., Ki67, Sca1, Sma, Pcam1, CD44, Lyve1, and EpCam (epithelial) (Fig. 5e–h). Again, these proliferative, exocytic stromal cells were mostly enriched in NPE limb (cell cluster 7) and trunk RMS (Fig. 5e–h). Curiously, a small percentage of endothelial cells with progenitor features (Pcam1^pos, Sca1^pos) was almost exclusively

present in NPE tumors (cell cluster 22 in limb and 15 in trunk) (Fig. 5e–h). Expression levels of markers defining tumor and stromal cells were also assessed by tSNE heatmaps that demonstrated the higher signal intensity of some of the markers (Des, Sma, Sca, Pcam1, Lamp1, and Ki67), specifically in NPE RMS (Supplementary Fig. 7a, b). These tumors also contained a higher percentage of CD45^pos macrophages, T cells, and B cells, than PE tumors (Supplementary Fig. 7c–f), likely contributing to an inflammatory pro-tumorigenic microenvironment. However, a more detailed analysis of the immune infiltration would be required to tease out its function in these tumor models. These results, based on single-cell protein expression, suggest that in NPE tumors, not only cancer cells but also stromal cells are maintained in an intermediate state of transdifferentiation that could be the basis of their distinct pleomorphic phenotype.

**Pleomorphic Neu1⁺/⁻/Ptch1⁺/⁻/ETV7^TG+/⁻ RMS develop an adipocytic metabolic signature.** Although the cell composition of NPE and PE limb and trunk tumors was similar, there was a distinctly higher percentage of cell populations expressing specific combinations of markers in NPE vs PE tumors (Fig. 5e, f). Taking the latter into account, we queried microarray data from individual NPE and PE limb and trunk RMS in search of distinct gene expression signatures that would add to their respective protein expression profiles and support pleomorphism. The microarray analysis allowed for the detailed identification of differentially expressed genes between NPE and PE tumors. The volcano plot and heatmap (Fig. 6a, Supplementary Fig. 8a) comprising a cohort of limb and trunk RMS showed that, compared with PE, NPE tumors had 491 upregulated and 287 downregulated genes (Fig. 6a, Supplementary Fig. 8a, Supplementary Data 3). Using Enrichr[51,52], we found that most of the upregulated genes in NPE RMS belong to pathways of lipid metabolism, myogenesis, and epithelial-to-mesenchymal transition (EMT) (Supplementary Fig. 8b, Supplementary Data 4), whereas genes downregulated in NPE belong to the following pathways: matrix metalloproteinases, cytokine-cytokine receptor interaction, HIF-1 signaling pathway and lung fibrosis (Supplementary Fig. 8c, Supplementary Data 5). GSEA (geneset enrichment analysis) of the total genetic landscape of limb and trunk RMS from NPE and PE mice identified the following geneset libraries: the KEGG PPAR (peroxisome proliferator-activated receptor), the Gene Ontology (GO) fatty acid metabolic process, the GO brown fat cell differentiation, and the KEGG vascular smooth muscle contraction (Fig. 6b). The leading edge of all four libraries/pathways positively correlated with NPE upregulated genes (Fig. 6b). White, beige and brown adipogenesis, and fatty acid metabolism are regulated by the essential transcription factor PPARγ[53]. In three out of the four aforementioned GSEA pathways, *Pparg* was one of the most upregulated genes in NPE RMS (Fig. 6b, Supplementary Data 6), together with some of its direct target genes, e.g., *Fabp4*, (fatty-acid-binding protein 4 or adipocyte protein 2), *Cd36* (cluster of differentiation 36), *AdipoQ* (adiponectin), and its receptor *AdipoR1*, but not *AdipoR2* (Fig. 6a, c, Supplementary Data 6).

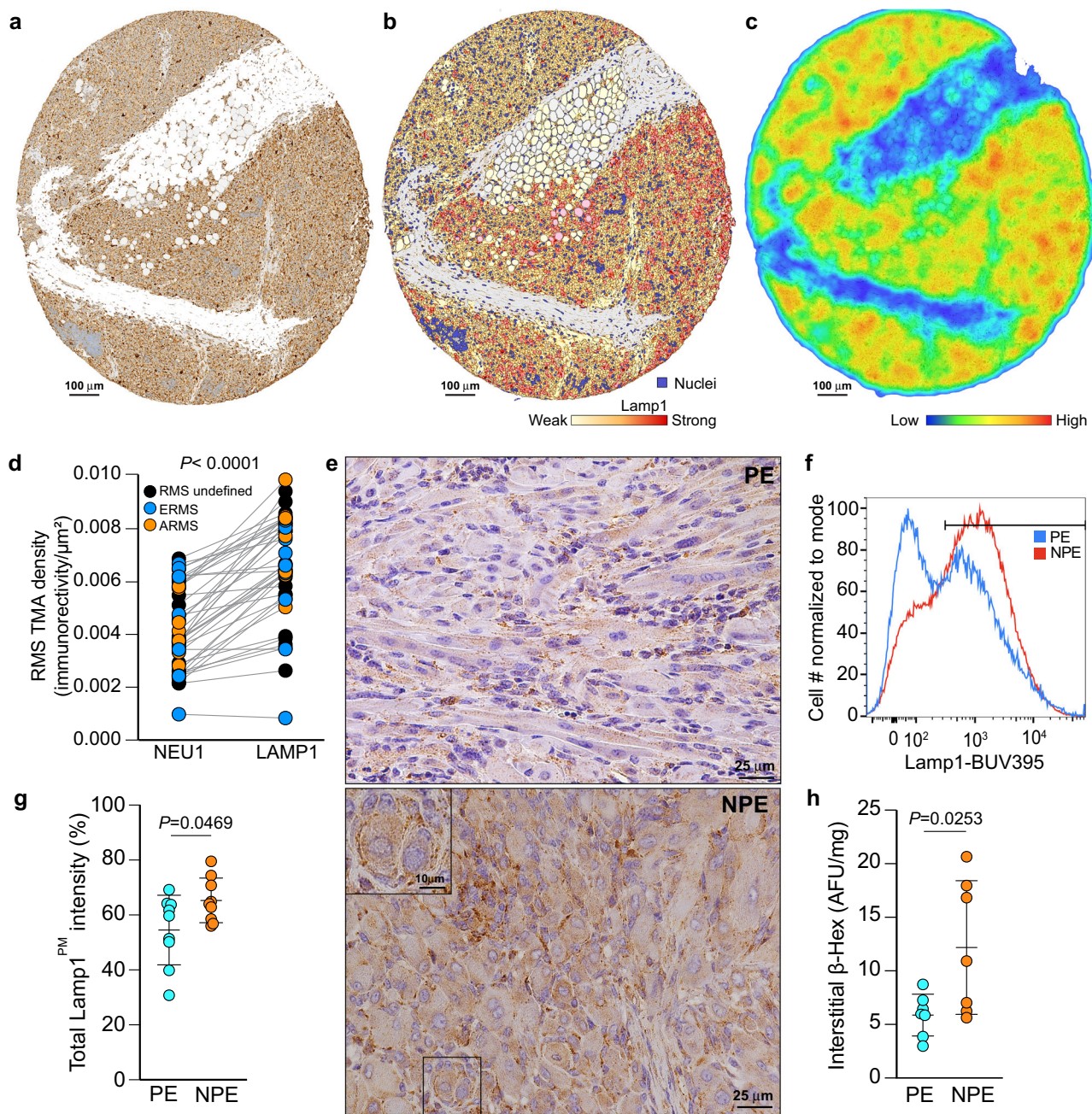

**Fig. 3 LAMP1 redistribution at the plasma membrane is a readout for increased lysosomal exocytosis downstream of low NEU1 expression.**
**a** Representative core of human RMS TMAs immunostained for LAMP1. Scale bar: 100 μm. **b** Density map of algorithm applied to a LAMP1-stained core showing LAMP1$^{PM}$ minimal and maximum density and location. Scale bar: 100 μm. **c** Heatmap representing the intensity of LAMP1$^{PM}$ staining of the core shown in a. Scale bar: 100 μm. **d** Correlation between low NEU1 and high LAMP1$^{PM}$ immunoreactivity in paired cores of human RMS TMAs. Two TMAs were used; Paired $t$-test; $n = 41$ biologically independent samples. **e** Representative micrographs of increased Lamp1 immunostaining in NPE compared with PE RMS. Punctuated lysosomal staining, typical for Lamp1. Scale bar: 25 μm, inset scale bar: 10 μm. **f** Flow cytometry histogram showing higher Lamp1$^{PMpos}$ expression in cells in NPE RMS versus PE RMS. **g** Increased Lamp1$^{PM}$ intensity quantified by flow cytometry in NPE versus PE RMS. Mean ± s.d.; Paired $t$-test; $n = 9$ biologically independent tumors. **h** Increased β-hexosaminidase activity in the interstitial fluid from NPE versus PE RMS. Mean ± s.d.; Student's (unpaired) $t$-test; $n = 7$ biologically independent tumors.

The derivation of brown adipocytes from skeletal muscle precursors or beige adipocytes from white adipocytes is orchestrated by PPARγ recruitment of PRDM16 (PR (PRD1-BF1_RIZ1 homologous)-domain-containing 16), which together form a core transcriptional complex[53]. PPARγ also recruits EBF2 (the early B-cell factor 2), a selective determinant of brown and beige adipocyte precursors[54], and coactivates the expression of *UCP1* (uncoupling protein 1 of brown adipocytes), *PPARA,* and

*PRDM16*[53]. We found that in NPE RMS *Pparg*, *Prdm16,* and *Ebf2* were upregulated together with their target, *Ucp1* (Fig. 6c, Supplementary Data 6). It is well documented that beige adipocytes that transdifferentiate from white adipocytes are derived from mesodermal stem cells expressing *Pdgfra*, *Pdgfrb*, *Acta2,* and *Myh11*, whereas brown adipocytes share with skeletal muscle cells the combined markers of dermomyotome precursors, *Myf5* and *Pax7*[55,56]. In our NPE RMS, only *Acta2 and Myh11* were

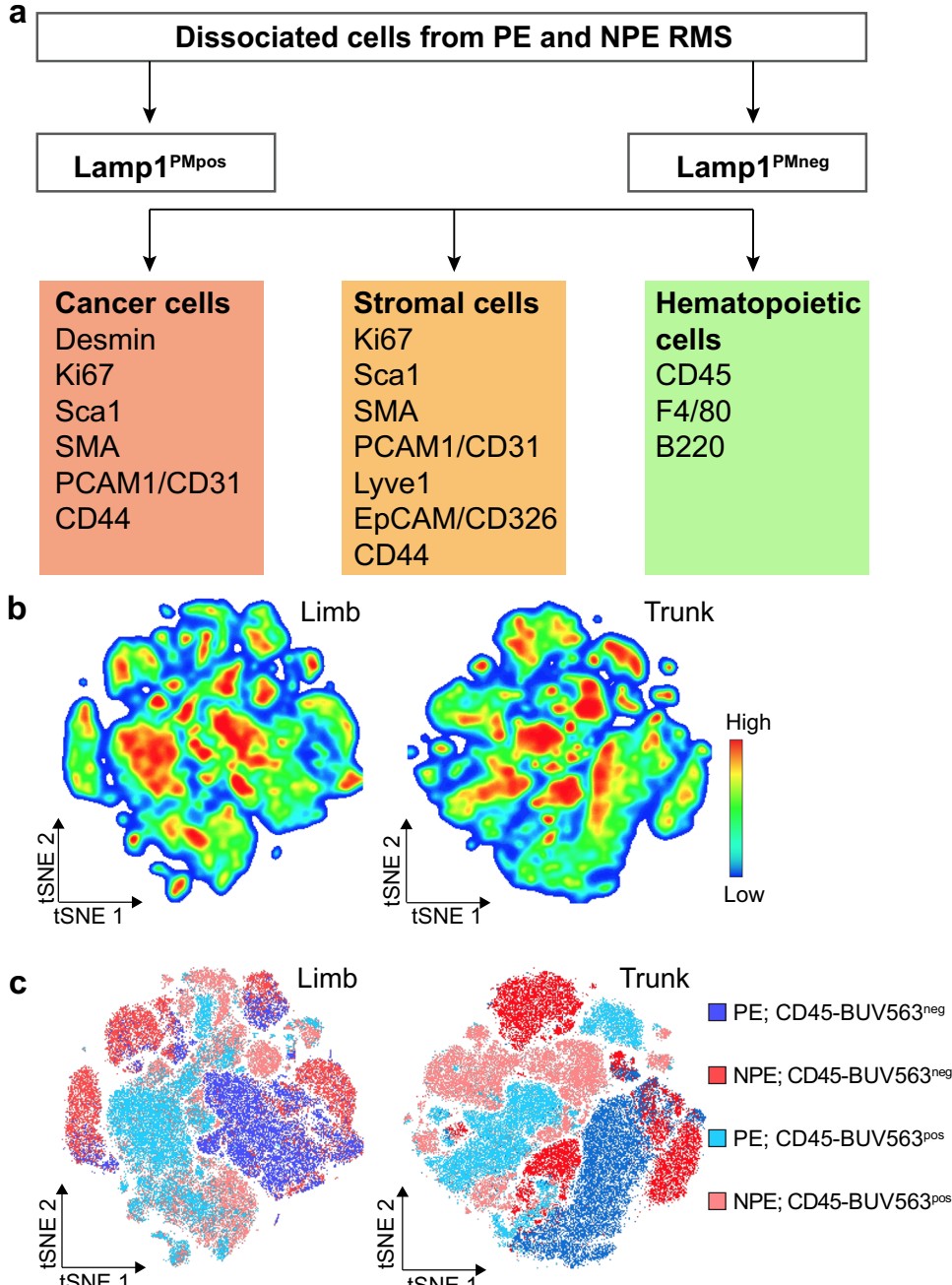

**Fig. 4 Single-cell multiplex flow cytometry analysis identifies murine RMS cell heterogeneity. a** Flow chart of multiplex flow cytometry analysis used to identify different cell populations in NPE and PE RMS. **b** tSNE density heatmaps of limb and trunk RMS from PE and NPE mice depicting areas of low and high cell number, based on the expression of the different markers (in **a**). **c** tSNE graphs of limb and trunk RMS from PE and NPE mice indicating the distribution of CD45[neg] and CD45[pos] cells.

significantly upregulated but not *Pdgfra, Pdgfrb, Myf5,* or *Pax7* (Fig. 6d, Supplementary Data 6), suggesting a preferential presence of white and beige adipocytes.

The earliest progenitor marker of myogenesis, *Pax3*[57] was downregulated in NPE vs PE RMS, while *Pax7* and *Myf5* showed no significant difference in expression (Fig. 6d). The other upstream regulators of myogenesis, *MyoD* and *Myog*, were significantly increased in NPE RMS (Fig. 6d), while the terminal differentiation gene *Des* was downregulated (Fig. 6d). Additionally, *Myocd* and *Myh11*, both smooth muscle specification genes, were highly upregulated in the NPE compared to PE RMS (Fig. 6d, Supplementary Data 6); *Myh11* was also found in the

leading edge of the KEGG vascular smooth muscle contraction pathway (Fig. 6b, Supplementary Data 6). Notably, human MYH11 protein, which is normally not expressed in RMS, is a marker of sarcoma pleomorphism[58,59], and an effector of lysosomal exocytosis downstream of *Neu1* downregulation[19]. So far, these results suggest that NPE RMS are already committed to a myogenic fate without reaching full differentiation. Moreover, the upregulation *Myh11* reinforces the notion that these tumors are both pleomorphic and exocytic.

Although NPE and PE RMS depend on the combination of *Ptch1*[+/−] and *ETV7*[TG+/−] for tumor specification and growth, the addition of *Neu1* haploinsufficiency drastically increased *Myc*

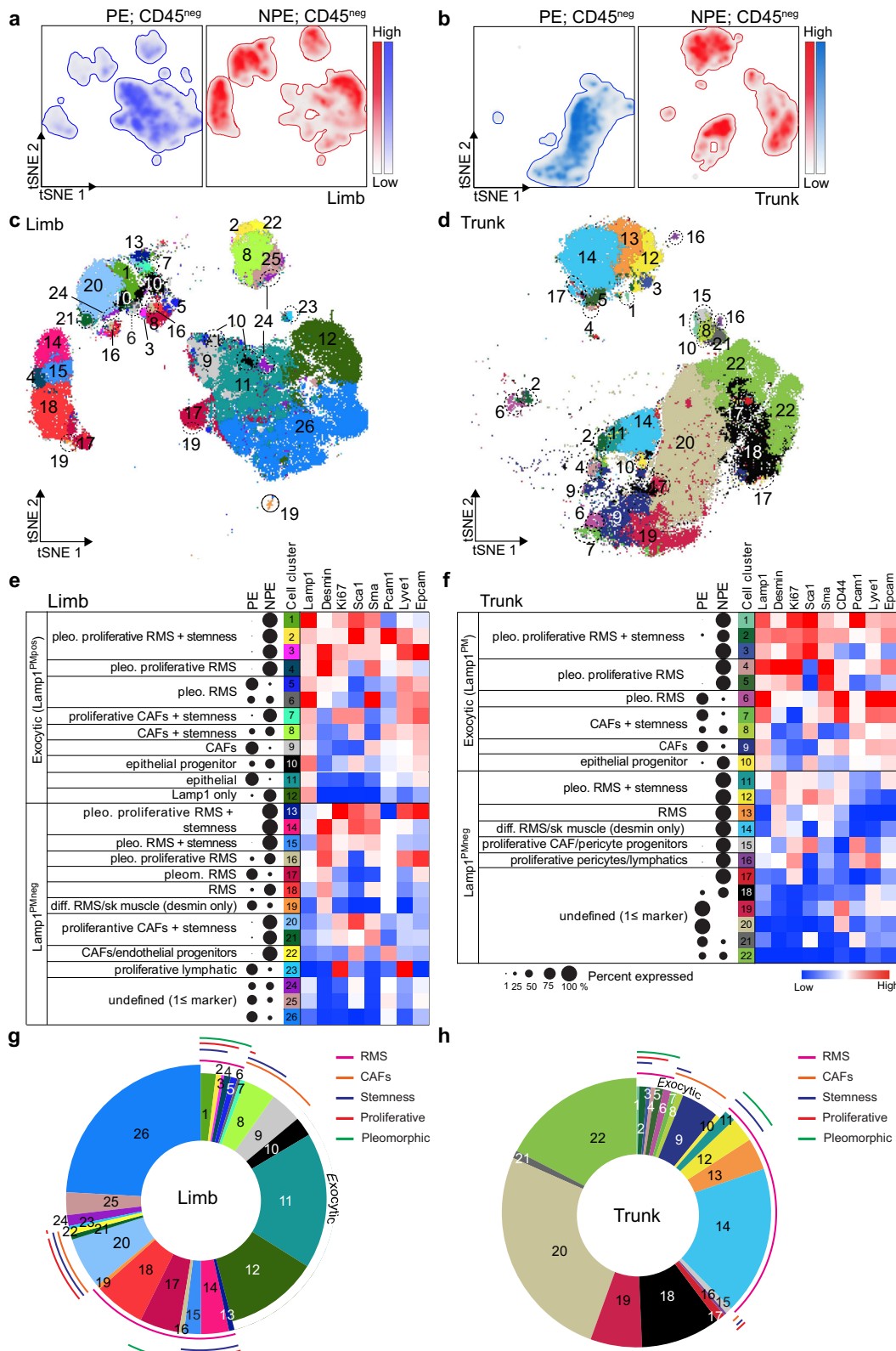

**Fig. 5 Lamp1$^{PMpos}$ cells are enriched in exocytic undifferentiated NPE RMS. a, b** tSNE graphs showing density and distribution of CD45$^{neg}$ cells from limb (**a**) and trunk (**b**) RMS of PE (blue) and NPE (red) mice. **c, d** tSNE of single-cell analysis of limb (**c**) and trunk (**d**) representation of the distinct cell clusters identified in RMS (numbered and colored) with the markers used with CD45$^{neg}$ cells from PE and NPE mice. **e, f** Annotated limb (**e**) and trunk (**f**) RMS CD45$^{neg}$ exocytic Lamp1$^{PMpos}$ and Lamp1$^{PMneg}$ cell populations from PE and NPE mice. Bubble map indicates the percentages of cells from PE and NPE contributing to a cell cluster. Heatmap shows the signal intensity of expression of the different markers used. **g, h** Pie graph of CD45$^{neg}$ cells from limb (**g**) and trunk (**h**) PE and NPE RMS showing the percentage of each cell cluster (numbered and colored matched).

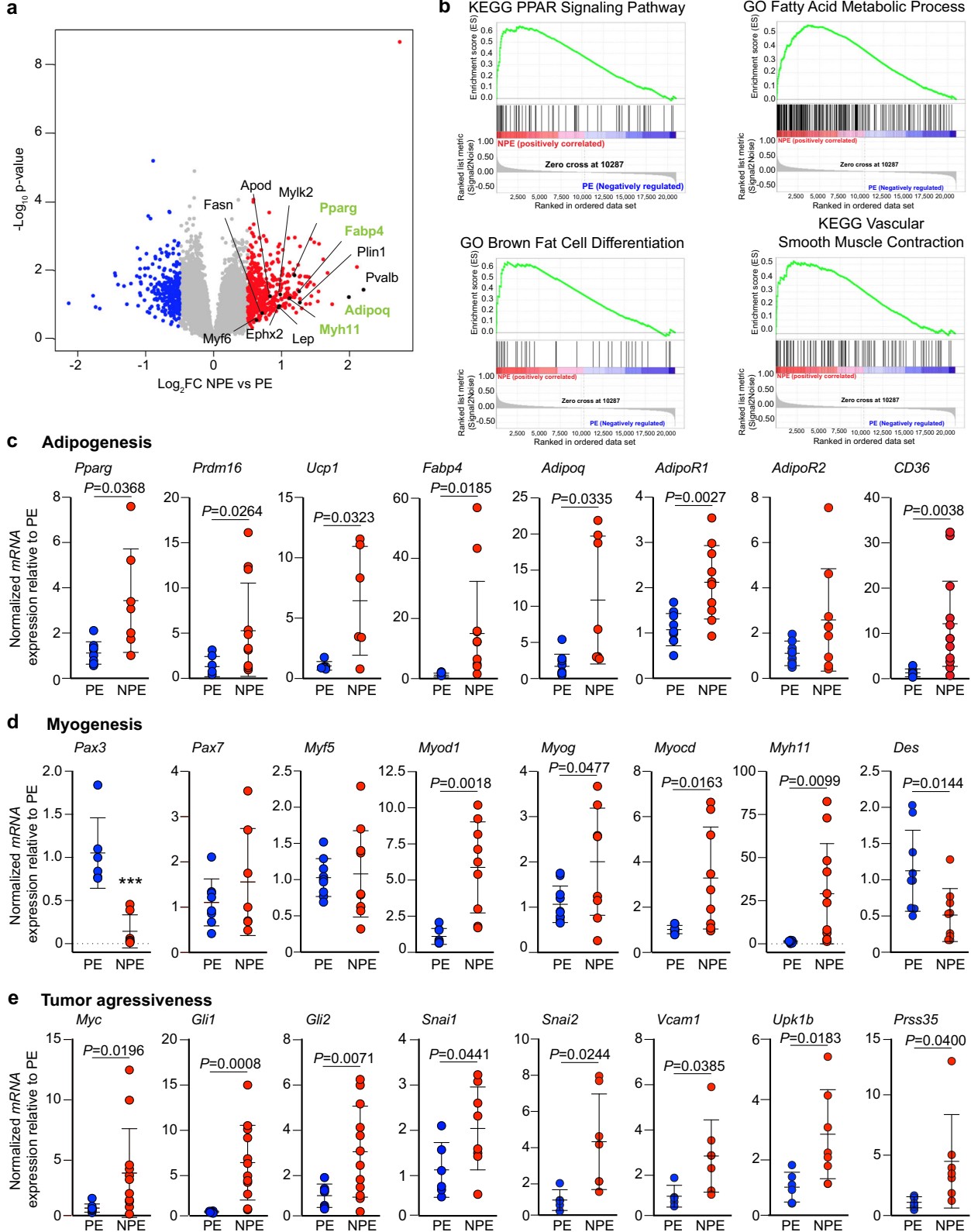

expression, the marker of proliferation, and the downstream effectors of the SHH pathway, *Gli1* and *Gli2*, known to be associated with metastatic growth (Fig. 6e). In agreement with the status of partial differentiation and smooth muscle/mesenchymal characteristics of NPE RMS, genes involved in epithelial-to-mesenchymal transition (EMT) and metastasis i.e., *Snai1, Snai2, Vcam1, Upk1b*, and *Prss35*, were also significantly upregulated

(Fig. 6e). It is noteworthy that the top-upregulated gene in NPE vs PE RMS is *H2Eb1*, encoding the mouse histocompatibility class II antigen E beta (MCH II), which is involved in antigen presentation, CD4+ T cell recognition and activation of the immune response[60] (Fig. 6a). The relevance of this observation will require extensive future study exploring the immune response component of these tumors, which is beyond the scope of this work.

**Fig. 6 Adipocytic metabolic genetic landscape defines pleomorphic NPE RMS. a** Volcano plot comparing differentially expressed genes in NPE versus PE RMS. Several genes enriched in the pathways identified by GSEA are annotated in the graph. Genes annotated in green were confirmed to be highly increased by qRT-PCR analysis. **b** Main KEGG and GO pathways identified by GSEA of differentially expressed genes in NPE vs PE RMS. FDR < 0.25. **c–e** Tumor qRT-PCR analysis showing features of adipogenesis (**c**); *Pparg* (PE: $n = 10$; NPE: $n = 7$), *Prdm16* and *Fabp4* (PE: $n = 7$; NPE: $n = 12$), *Ucp1* ($n = 6$), *Adipoq* (PE: $n = 9$; NPE: $n = 7$), *AdipoR1* ($n = 10$), *AdipoR2* (PE: $n = 10$; NPE: $n = 9$), *CD36* (PE: $n = 6$; NPE: $n = 11$), myogenesis (**d**); *Pax3* (PE: $n = 6$; NPE: $n = 7$), *Pax7* (PE: $n = 8$; NPE: $n = 7$), *Myf5* (PE: $n = 8$; NPE: $n = 6$), *Myod1* and *Myocd* (PE: $n = 8$; NPE: $n = 9$), *Myog* (PE: $n = 10$; NPE: $n = 9$), *Myh11* (PE: $n = 8$; NPE: $n = 11$), *Des* ($n = 9$) and tumor aggressiveness (**e**); *Myc* (PE: $n = 9$; NPE: $n = 12$), *Gli1* and *Gli2* (PE: $n = 7$; NPE: $n = 12$), *Snai1* (PE: $n = 6$; NPE: $n = 8$), *Snai2* and *Vacm1* (PE: $n = 5$; NPE: $n = 7$), *Upk1b* (PE: $n = 6$; NPE: $n = 7$), *Prss35* ($n = 7$) in NPE versus PE RMS. qRT-PCR results are from normalized mRNA expression relative to PE RMS samples. Mean ± s.d.; Welch (unpaired) *t*-test; biologically independent tumors.

To exclude the possibility that *Neu1* haploinsufficiency by itself can cause the significant changes in gene expression observed between PE and NPE tumors, we performed RT-qPCR analysis of the same set of genes in normal muscle tissue from both NPE and PE mice and found no differences (Supplementary Fig. 9a–c). Overall, our results indicate that NPE RMS maintains a status of intermediate differentiation and pleomorphism fueled by a strong adipogenic component, likely linked to *Neu1* haploinsufficiency.

**Pleomorphic *Neu1*$^{+/-}$/*Ptch1*$^{+/-}$/*ETV7*$^{TG+/-}$ RMS share an adipogenic signature with human ARMS and ERMS.** Based on these data, we revisited the H&E and Masson's trichrome-stained PE and NPE RMS and identified a slight increase in the percentage of adipose cells in the NPE sections, although this difference was not statistically significant (Supplementary Fig. 10a). These were especially enriched in the pleomorphic foci and the invasive fronts, showing extensive collagen deposition and ECM remodeling (Fig. 1h asterisk, Fig. 2b). To confirm the gene expression data, we probed NPE and PE tumor sections with adiponectin the product of the *AdipoQ* gene that was highly upregulated in the NPE tumors and compared its expression pattern with that in human RMS TMAs (Fig. 7a, b). Adiponectin, the essential secreted adipokine in adipocyte signaling, was highly expressed in NPE tumors compared to PE, and was detected mostly at the cell surface of adipocytes and tumor cells, including large multinucleated pleomorphic cells (Fig. 7a). In TMAs from RMS patients, adiponectin immunoreactivity was positive in all cores, albeit of low intensity, and localized to the cytoplasm of neoplastic cells. However, in a few cores, adiponectin intensity was moderate to high (Fig. 7b). Notably, white adipose tissue was detected in 25% (10/40) of RMS cores. Querying the human RNAseq PCGP dataset, we found a group of five RMS patients with high expression of *ADIPOQ*, that was clearly above the standard deviation of the median expression of all analyzed samples ($n = 44$) (Fig. 7c). These patients were all diagnosed with ARMS, the highest expression was from recurrent disease, and one had an undefined RMS subtype (Fig. 7c). All five patients with *ADIPOQ* upregulation also showed higher than median levels of *PPARG*, *CD36,* and *MYH11*, but expressed lower levels of *UCP1* (Fig. 7c–g). Most importantly, we found that ARMS patients expressed significantly higher levels of *ADIPOQ*, *PPARG* and *MYH11* when compared with ERMS patients (Fig. 7h–l). Using GSEA we also compared *ADIPOQ* expression in RMS patients with geneset profiles from the MSigDB collection (Supplementary Fig. 10b). This analysis confirmed that the RMS patients with the highest *ADIPOQ* expression also showed upregulation of genes belonging to the pathways of adipogenesis, vascular smooth muscle contraction, and ECM receptor interaction, but not myogenesis (Supplementary Fig. 10b, Supplementary Data 7). These results support the presence of an adipose component associated with undifferentiated characteristics in some of the aggressive forms of human RMS.

## Discussion

Pleomorphic RMS is a rare and aggressive form of sarcoma, occurring mostly in adults that is often difficult to diagnose and treat[4]. These tumors present with a mixture of large, atypical pleomorphic rhabdomyoblasts, often multinucleated, and partially differentiated spindle-like cells[35].

By lowering the expression levels of the lysosomal sialidase *Neu1* in the *Ptch1*$^{+/-}$/*ETV7*$^{TG+/-}$ genetic background, we have successfully developed a spontaneously occurring mouse model of pleomorphic RMS without the use of conditional alleles. It is important to emphasize that the effect of *Neu1*$^{+/-}$ is not in the generation of RMS, but rather in its pleomorphic transformation once the tumor is initiated; while ETV7 expression is promoting a high incidence of RMS formation in *Ptch1*$^{+/-}$ mice[32]. Although NPE tumors arise preferentially in the limbs and trunk and only express 10% of the genes that trended with human ARMS, their genetic features are much closer to ERMS. Unfortunately, the unavailability of compatible genetic data from human pleomorphic RMS that we could compare to our NPE datasets, limits our current evaluation of NPE tumors to their histological characteristics, and further investigation is required to confirm NPE mice as a model of human pleomorphic RMS.

Exploiting the low NEU1 > high lysosomal exocytosis axis, we developed a single-cell approach that relied on the increased levels of the Neu1 substrate Lamp1 at the PM of cancer and stromal cells. This approach allowed us to pinpoint tumor heterogeneity, consisting of specific cell populations that are highly exocytic and likely responsible for initiating/perpetuating RMS transformation and ECM remodeling. The ensuing fibrotic microenvironment surrounding pleomorphic foci in NPE RMS is typical of the most aggressive and chemo-resistant tumors in humans[61].

The presence of mature adipocytes together with adipose-like cells in both murine and human RMS could be explained by a shared developmental ancestry between skeletal muscle and brown fat[62,63]. However, brown adipocytes originate from early dermomyotomes (*Pax7*$^+$/*En1*$^+$/*Myf5*$^+$), while distinct somite populations expressing *MyoD* downstream of *Myf5* are already committed to myogenesis. Therefore, fate determination between brown adipocytes and committed somite populations should happen prior to *MyoD* expression[56]. Given the high levels of *MyoD* detected in NPE RMS, we can infer that these tumors have reduced brown adipocytes. In contrast, they might be enriched in beige adipocytes, since these derive from progenitors expressing Sma, Myh11, Pdgfra, or Pdgfrb, and transdifferentiate from mature white adipocytes stimulated, among others, by PPARγ activation and release of adipokines, including adiponectin[56,64,65]. In this scenario, tumor cells, cancer-associated fibroblasts, and cancer-associated adipocytes may together sustain cancer progression and chemo-resistance by fueling metabolism and maintaining a pro-fibrotic microenvironment downstream of low NEU1 activity. These combined features, largely depending on posttranslational modifications of glycans, appear to be common between our NPE model and human RMS, given that increased

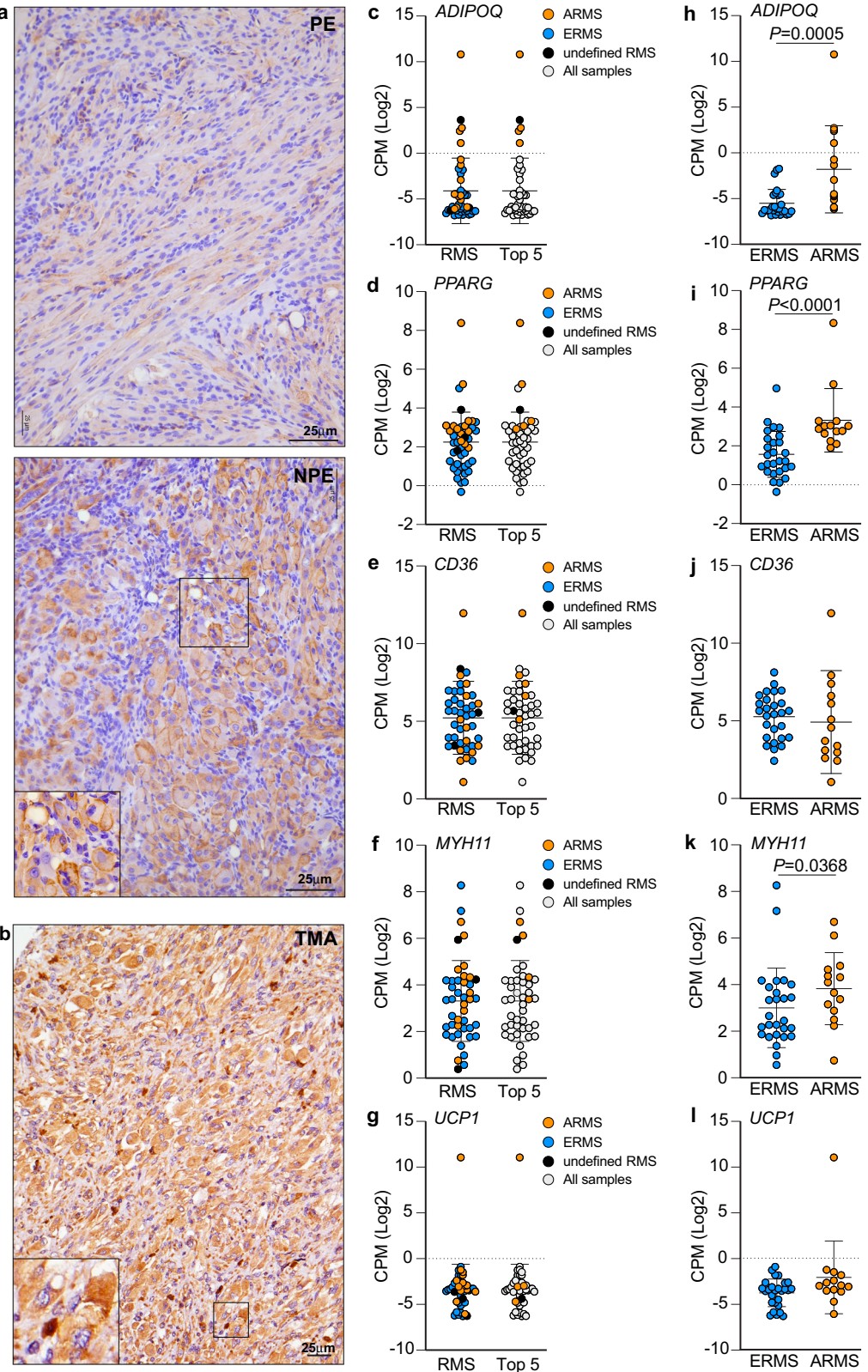

adipocytes have been detected in poorly differentiated, human metastatic or recurrent RMS, as well as in chemotherapy-treated RMS[66–68].

We propose that the NPE mice can be exploited as a pre-clinical model of pleomorphic RMS to understand the pathology of these aggressive tumors and explore new therapeutic opportunities in pathways controlled by NEU1. Furthermore, the overlooked adipose component in these tumors underscores the importance of combining the use of adipose markers during the diagnosis of this type of RMS.

## Methods

**Mice**. All procedures were performed following NIH guidelines and animal protocols approved by the St Jude Children's Research Hospital Institutional Animal

**Fig. 7 Adipogenic features of NPE RMS are shared with human RMS. a** Representative micrographs of increased adiponectin immunostaining in NPE versus PE RMS. Increased cell surface positive staining is depicted in the zoom inset of NPE RMS. Scale bar: 25 μm. **b** Representative micrograph of strong adiponectin immunostaining in a core of an undifferentiated human RMS from 2 TMAs; $n = 40$ biologically independent samples. Intense staining is depicted in the zoom inset. Scale bar: 25 μm. **c–g** Representation of genetic expression of *ADIPOQ* (**c**), *PPARG* (**d**), *CD36* (**e**), *MYH11* (**f**), and *UCP1* (**g**) across human patient RMS RNAseq data (PCGP). Gene expression is represented in relation to the mean value. All samples of ERMS (blue), ARMS (orange) and undefined RMS (black) analyzed are shown first on the x-axis (RMS), followed by the representation of the cluster of the top five samples with the highest *ADIPOQ* expression (Top 5). $n = 44$ biologically independent samples. **h–l** Comparison of *ADIPOQ* (**h**), *PPARG* (**i**), *CD36* (**j**), *MYH11* (**k**), and *UCP1* (**l**) expression between ERMS and ARMS patients. Mean ± s.d; Mann–Whitney t-test used to compare ERMS and ARMS; $n = 27$ and $n = 14$ biologically independent samples, respectively.

Care and Use Committee (IACUC); animal protocol #235–100636. Neu1[+/−] FVB/ NJ[69] mice were crossed with Ptch1[+/−]/ETV7[TG/+/−] (129sv/C57BL/6) mice[32,33]. The mice, both males, and females (1 month–24 months), used in this study were obtained (over a timespan of 2 years) from the same strategic breeding schedule: Neu1[+/−]/Ptch1[+/−]/ETV7[TG/+/−] X Neu1[+/−]/Ptch1[+/−]/ETV7[TG+/+], but PE and NPE mice with tumors not necessarily came from the same litter. Mice presenting with masses were observed for lethargy and sacrificed when moribund, or if they displayed symptoms such as paralysis or when the tumor volume reached 20% of body mass in accordance with IACUC-approved guidelines.

**Real-time quantitative polymerase chain reaction**. Total RNA was isolated from mouse tumors using the PureLink RNA Kit (Life Technologies), and DNA contaminants were removed on a deoxyribonuclease I column (Life Technologies), according to the manufacturer's protocol. RNA quantity and purity were measured using a NanoDrop Lite spectrophotometer (Thermo Fisher Scientific). Complementary DNA was synthesized using 0.5–5 μg of total RNA with RT2 First Strand Kit (QIAGEN). RT-qPCR was performed using RT2 SYBR Green qPCR Mastermix, 1 μl (8, 16, or 50 ng) of complementary DNA, 10 μM primer, and ribonuclease-free water in a 25 μl reaction volume on an iQ5 or CFX96 real-time PCR machine (Bio-Rad). Samples were normalized to 18 S ribosomal RNA. The specific primers used are summarized in Supplementary Table 1.

**Western blotting and co-immunoprecipitations**. Frozen tumors were powdered by grinding in liquid nitrogen and homogenized in 1× Cell Signaling lysis buffer [20 mM tris-HCl, (pH 7.5), 150 mM NaCl, 1 mM Na2EDTA, 1 mM EGTA, 1% Triton X-100, 2.5 mM sodium pyrophosphate, 1 mM β-glycerophosphate, 1 mM Na₃VO₄, and leupeptin (1 μg/ml) (Cell Signaling Technologies) supplemented with 1 mM phenylmethylsulfonyl fluoride (PMSF), for 2 min at 30 Hz in the Omni Prep Multi-Sample homogenizer (QIAGEN). Lysates were spun through a QIAshredder column (Qiagen), and freeze-thawed three times. After centrifugation at $20,000 \times g$ for 30 min at 4 °C, the protein concentration in the supernatant was determined using a BCA protein assay kit (Bio-Rad). Per sample, 750 μg of protein was subjected to ETV7 IP. Antibody (2 μg) was added to 500 μl of lysate, and samples were rotated overnight at 4 °C. Protein G–coated Dynabeads (Life Technologies) was added (10 μl per sample), and samples were rotated at 4 °C for 90 min. Bead-antibody-protein complexes were captured using a DynaMag-2 magnet (Life Technologies) and washed four times with CHAPS lysis buffer [40 mM Hepes (pH 7.4), 1 mM EDTA, 120 mM NaCl, 10 mM sodium pyrophosphate, 10 mM β-glycerophosphate, 0.3% CHAPS, 50 mM NaF, 1.5 mM NaVO, 1 mM PMSF, and one tablet of EDTA-free protease inhibitors (Roche) per 10 ml solution]. Antibody-protein complexes were retrieved from the beads by heating at 70 °C in 1.25× LDS loading buffer (Life Technologies) in CHAPS lysis buffer and loaded on precast 4–12% bis-tris protein gels (Life Technologies). Proteins were transferred onto nitrocellulose membranes using the iBLOT system (Life Technologies) following the manufacturer's protocol. Membranes were blocked with 5% milk and 0.1% Tween 20 in tris-buffered saline (TBS) and incubated with the appropriate antibodies (Supplementary Table 2) in 5% bovine serum albumin in TBS with 0.1% Tween 20 overnight at 4 °C. All primary antibody incubations were followed by incubation with secondary horseradish peroxidase (HRP)–conjugated antibody (Pierce) in 5% milk and 0.1% Tween 20 in TBS and visualized with SuperSignal West Pico or Femto Chemiluminescent Substrate (ThermoScientific), using a Biorad ChemiDoc MP imaging system.

**Hematoxylin & Eosin (H&E) staining and Immunohistochemistry (IHC)**. Murine tumor tissues were fixed in 10% buffered formalin and embedded in paraffin. Sections of 6 μm were cut and deparaffinized. H&E staining was performed following standard procedures. For immunohistochemistry, after deparaffinization, antigen retrieval was performed using citrate buffer [10 mM Tris-sodium citrate, pH 6.0, 0.05% Tween 20] (for Neu1 and LAMP1), or Tris-EDTA [10 mM Tris Base, 1 mM EDTA pH9, 0.05% Tween 20] (for MyoD, Myogenin and Adiponectin). Endogenous peroxidase was removed by incubation with 3% hydrogen peroxidase in methanol for 5 min. For Neu1, Lamp1, and adiponectin, we used the ImmPress HRP polymer system following the manufacturer's instructions (VECTOR Laboratories cat# MP-7801) with overnight incubations with primary antibodies (Supplementary Table 2). Mouse anti-MyoD and anti-

Myogenin antibodies were biotinylated with the ARK (Animal Research Kit), Peroxidase (DAKO K3954) following the manufacturer's instructions. Signal detection was performed with the Vectastain Elite ABC HRP (Vector Laboratories Cat# PK6100) followed by Stable DAB chromogen (Invitrogen Cat#75018). Slides were counterstained with hematoxylin, dehydrated, and mounted with a xylene-based mounting medium. Formalin-fixed paraffin-embedded human RMS TMAs were obtained from the St Jude Pathology/Laboratory Medicine Department. Specimens were de-identified, and the study was approved by the Institutional Review Board (IRB approval Pro00008511) as non-human subject research.

Human RMS TMAs were obtained from the St Jude Pathology/Laboratory Medicine Department (approved by IRB Pro00008511).

**Masson's Trichrome staining**. Masson's Trichrome staining was done as previously described[70]. In brief, FFPE (formalin-fixed paraffin-embedded) murine RMS samples and patient TMAs on slides were fixed in Bouin's solution for 1 h at 60 °C. Sections were then stained sequentially at room temperature with Weigert's iron hematoxylin, Biebrich scarlet-acid fuchsin, phosphotungstic/phosphomolybdic acid, and aniline blue. Sections were washed, dehydrated, and mounted with a xylene-based mounting medium. Trichrome Masson's-stained slides were scanned to ×20 scalable images with an Aperio Scanscope XT (Leica Biosystems, Inc.). The Aperio color deconvolution and colocalization algorithms and the ImageScope software v12.4.3 (Leica Biosystems, Inc.) were used to quantify ECM/collagen (blue) deposition in the microenvironment and within tumor cells.

**Nuclear morphometry analysis**. H&E-stained murine RMS samples were scanned to ×20 scalable images with an Aperio ScanScope XT (Leica Biosystems, Inc.) and annotated using ImageScope software v12.4.3 (Leica Biosystems, Inc.). The Genie tissue classifier was trained to identify tumor nuclei and quantify their numbers and sizes.

**Quantification of LAMP1 staining**. IHC stained images from RMS TMAs were scanned with a Pannoramic 250 Flash III (3DHistech, Inc.) to ×32 scalable images. Images from tissue cores from the whole slide were then manually annotated using HALO v3.2.1851.354 software (Indica Labs). A Membrane v1.7 algorithm was trained to identify LAMP1 immunoreactivity at the membrane level. After positive immunoreactivity was identified with the algorithm, a density heatmap spatial analysis was performed on the analyzed samples to determine the minimal and maximal intercellular density and spacing of immune-positive neoplastic cells within a 25 μm radius in each core using the Spatial Analysis Module HALO 3.2 (Indica labs).

**β-Hexosaminidase activity in tumor's interstitial fluid**. Frozen tumors were powdered by grinding in liquid nitrogen and homogenized in PBS for 2 min at 30 Hz in the Omni Prep Multi-Sample homogenizer (QIAGEN). Interstitial fluid was purified by sequential centrifugation steps at $300 \times g$ for 10 min, $2000 \times g$ for 10 min, and $10,000 \times g$ for 30 min to remove cells and cell debris. The supernatant/interstitial fluid was spun in an ultracentrifuge at $100,000 \times g$ for 2 h (SW32Ti rotor) to remove small vesicles and exosomes. All steps were performed at 4 °C. Interstitial fluid was diluted 5 times in PBS and spun through a Sephadex column at pH 5.5 for 2 min at $850 \times g$. Beta-hexosaminidase activity[19] was measured with 4-Methylumbelliferyl N-acetyl-β-D-glucosaminide fluorometric substrate (3 mmol/L in citrate phosphate buffer pH4.4 (6 mmol/L citric acid, 10 mmol/L disodium hydrogen phosphate) (Sigma M2133). Ten μL of interstitial fluid, or when needed a 1:10 dilution, was incubated at 37 °C with 10 uL of substrate for 1 h. The reaction was stopped with 200 mL carbonate stop buffer (0.5 M Na₂CO₃ with the pH set to 10.7 by adding 0.5 M NaHCO₃). The amount of fluorescence generated from each reaction indicates cleavage of the fluorescent tag from its peptide substrate and is proportional to the amount of enzyme present. The mean fluorescence of samples given assay buffer was subtracted from the mean fluorescence of samples given assay buffer plus the specific substrate. Mean fluorescence values were interpolated to a standard curve, adjusted for dilution and length of incubation, and normalized to protein by BCA. The final units of activity were reported as nanomoles per hour per milligram.

**Gene expression analyses**. For microarray analysis of NPE and PE trunk and limb RMS, total RNA (100 ng) was converted into biotin-labeled cRNA (Ambion WT Expression Kit, Affymetrix Inc) and hybridized to Clariom S Mouse GeneChip (Affymetrix Inc) and signals summarized by RMA (Affymetrix Expression Console v1.1). Probe signals from arrays were normalized and transformed into log2 transcript expression values using the Robust Multiarray Average algorithm (Partek Genomics Suite v6.6). Patient's RMS gene expression data from the Pediatric Cancer Genome Project (PCGP) were also used. For the comparison of murine versus human RMS, principal component analysis (PCA) and quality-control metrics removed outliers from both PCGP RNAseq and murine microarray data. We used data from 8 ARMS with PAX3/FOXO1 fusion, 4 ARMS with PAX7/FOXO1 fusion and 22 ERMS (accession# EGAS00001000256)[71]. Metadata were also supplemented for fusion calls from public RMS data (PeCan https://pecan.stjude.cloud/home). Correlations were calculated with Pearson's correlation of deciles of the mean log2 FPKM (for RNAseq) and RMA values for the microarray's expression by class. There were 91 genes that passed the false discovery rate (FDR) at 5%, having a difference of 5 or more deciles. GSEA (https://www.gsea-msigdb.org/gsea/index.jsp)[72,73] was also performed using the curated pathways from MSigDB[72,74]. Differentially expressed transcripts were identified by ANOVA, and the FDR was estimated. Functional enrichment analysis of gene lists was performed using the DAVID bioinformatics databases (https://david.ncifcrf.gov/). Additional gene sets for transcriptional analysis were identified using Enrichr[51,52,75]. For adipogenic gene expression analysis, gene level feature counts were downloaded from the St. Jude PCGP for 44 patient samples of RMS (14 ARMS, 27 ERMS, and 3 RMS undefined). These counts were normalized, and log2 transformed using voom-limma, and then log2CPM was graphed in Prism. The normalized, transformed counts were also used for GSEA analysis using a combination of experimentally derived gene sets, as well as established gene sets from MSigDB[72,74]. Normalized RMA microarray values were compared using voom-limma in R and displayed in a volcano plot.

**Multiplex flow cytometry**. NPE and PE tumors were dissociated with the Mouse Dissociation kit and gentle MACS Octo Dissociator with heaters following the manufacturer's instructions (MACS, Miltenyi Biotec). Large aggregates persisting in the cell preparation were then removed by filtering them through 70 μm cell strainers. The resulting single-cell suspensions were stained with fluorochrome-conjugated monoclonal antibodies as noted in Supplementary Table 2, and analyzed using a BD FACSymphony A9 analyzer (BD, San Jose) equipped with 355, 400, 440, 488, 561, and 640 nm lasers for excitation and an array of 30 detectors with the appropriate light filters for resolving the specified fluorochromes. Self-organizing maps for visualizing and interpreting cytometry data, FlowSOM, was used[76] on 60,000 cells from limb and trunk NPE and PE RMS to generate density, heatmaps, and tSNE graphs. Cellular clusters were identified and annotated based on the differential expression of markers. Additional two-dimensional analyses were conducted using both FlowJo and FACS Diva software suites.

**Statistics and reproducibility**. Quantitative data are presented as mean ± s.d. of at least six biologically independent samples/tumors. A number of replicates are noted in the figure legends. Statistical analyses were performed using GraphPad Prism. Student's t-test for unpaired and paired (two-tailed), Welch's t-test for unpaired t-test, one-way ANOVA t-test, Mann–Whitney for unpaired (two-tailed) t-test, all tests have a 95% confidence interval, and log-rank test for trend was performed to ascertain statistical significance and are noted in the figure legends. Mean differences were considered significant when $P < 0.05$. Source data underlying the main figures are presented in Supplementary Data 1.

**Reporting summary**. Further information on research design is available in the Nature Research Reporting Summary linked to this article.

## Data availability

The data that support the findings of this study are available from the corresponding authors on reasonable request. Uncropped and unedited blot images are provided in Supplementary Fig. 2. Microarray data in this publication are deposited in NCBI's functional genomic data repository Gene Expression Omnibus (GEO) and are assecible through GEO series accession number GSE212378. Previously published datasets used in this study were deposited in the European Bioinformatics Institue (EMBL-EBI) and accessible through accession number EGAS00001000256. Source data underlying Fig. 1b, d, I, j, 2d–g, 3d, g, h, 6c–e, 7c–l and Supplementary Fig. 1a, 3a, b, d, e, 9a–c, 10a are provided within this paper (Supplementary Data 1). Patient's RMS gene expression data can be accessed through StJude open resource page (PeCan https://pecan.stjude.cloud/home).

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

## Acknowledgements

We thank I. Annunziata for the critical reading and input. A.d'A. holds the Jewelers for Children Endowed Chair in Genetics and Gene Therapy. G.C.G. holds the Albert and Rosemary Joseph Endowed Chair in Genetic Research. This work was funded in part by grants from the by NIH grants CA021764 and 5R01GM104981, CCSG St. Jude Developmental Funds, the Assisi Foundation of Memphis, and the American Lebanese Syrian Associated Charities (ALSAC).

## Author contributions

E.M. designed, performed, and analyzed all the experiments, and wrote the manuscript. D.v.d.V. performed all the RT-qPCRs, isolated interstitial fluid, did statistical analyses, analyzed data, assembled the figures, and edited the manuscript. H.T. diagnosed and analyzed all formalin-fixed, paraffin-embedded samples of mouse tumors and TMAs of RMS patients, and performed all computational algorithm analyses. S.P. helped design, performed, and analyzed all FACS and FlowSOM data. S.M.D. helped analyze microarray data, performed GSEA, and analyzed PCGP human RMS data and helped with the final figures pertaining to these analyses. J.L. and R.A. performed the FACS and FlowSOM analyses. D.F. performed correlation analyses of murine RMS vs human RMS genes. G.N. performed and analyzed the microarray data. H.H. embedded, processed, and cut sections of all mouse tumors. F.H. performed mTORC3/ETV7 co-immunoprecipitations. S.K. diagnosed TMAs of RMS patients. G.C.G. designed experiments, analyzed data, edited, and wrote the manuscript. A.d'A. conceived the project designed experiments, analyzed data, supervised the study, edited, and wrote the manuscript. All authors approved the final manuscript.

## Competing interests

The authors declare no competing interests.
