## [Peer Review File · Communications Biology]

Reviewers' comments:

Reviewer #1 (Remarks to the Author):

The authors of the manuscript "Modelling pleomorphic rhabdomyosarcoma in mice by haploinsufficiency of the 2 lysosomal sialidase NEU1" studied the effect of Neu1 haploinsufficiency in Ptch1+/-/ETV7TG/+/- (PE) mice that develop rhabdomyosarcoma (RMS) with high incidence. Neu1 is a lysosomal sialidase that negatively regulates exocytosis of lysosomes by cleaving the sialic acids of LAMP1. Thus, low NEU1 activity increases the number of lysosomes containing sialylated LAMP1 that dock at the plasma membrane. This finally results in excessive lysosomal exocytosis and leads to disruption of the integrity of the plasma membrane and extra cellular matrix.

The authors show that RMS in PE mice are different from those in Neu1+/- Ptch1+/-/ETV7TG/+/- (NPE) mice. Thus, whereas PE tumors are more differentiated, the RMS of NPE mice show cellular heterogeneity and numerous undifferentiated, anaplastic foci and are rich in pleomorphic and rhabdoid cells. In addition, connective tissue and intracytoplasmic collagen are elevated in NPE tumors suggesting that reduced Neu1 expression promotes pleomorphism and transforms the tumor stroma into a desmoplastic/fibrotic state.

The authors also show that low expression of NEU1 is accompanied by an overall increase in LAMP1 in murine (and also human) RMS. In addition, examination of b-hexosaminidase in the interstitial fluid of several NPE RMS confirmed that NPE tumors are more exocytic than PE RMS. Next, the authors performed multiplex flow cytometry. The data show that the tumors are composed of discrete cell clusters that differ not only between NPE and PE RMS, but also between limb and trunk tumors. Moreover, protein expression in selected CD45neg LAMP1pos cell populations suggest that cancer and stromal cells in NPE tumors are maintained in an intermediate state of transdifferentiation. Finally, GSEA analysis suggested that this intermediate differentiation and pleomorphism of NPE RMS have a strong adipogenic component.

The paper is very well written. All methods are described in detail and statistical analyses are adequate.

Although the study is predominantly descriptive, it contains interesting data.

However, some points need to be addressed:

- Inverse expression of Neu1 and LAMP1 has already been described in sarcoma including RMS. Similarly, it has been demonstrated that Neu1 haploinsufficiency fosters the development of pleomorphic sarcomas and that human RMS cells with low Neu1 and high LAMP1 show excessive lysosomal exocytosis (Machado et al. Sci. Adv. 2015;1:e1500603 18 December 2015). Therefore, the results shown in the current paper are not really new. The authors should comment on that.

- Neu1-/- mice show enhanced infiltration with connective tissue and alterations of the ECM (Zanoteli et al; Biochim Biophys Acta. 2010 ; 1802(7-8): 659-672. doi:10.1016/j.bbadis.2010.04.002.). Therefore, I am wondering whether normal muscle of Neu1+/- mice also shows these characteristics. If so these characteristics are not a RMS phenotype, but rather are related to the Neu1 mutation itself. Therefore, it is important to repeat the experiments shown in Fig.2a – Fig.2g on normal muscle tissue of NPE mice.

- Please provide an information about the genetic background of the mice and indicate whether the PE and NPE mice are littermates. This is a very important information, because the genetic background may influence some of the phenotypic characteristics of the tumors.

- Based on histology I agree with the authors that NPE tumors resemble human pleomorphic RMS. Therefore, I am wondering why the authors compared the gene/protein expression signatures of NPE tumors to that of human ARMS and ERMS but not to the signature of pleomorphic RMS. One example is the adipocytic signature. Were pleomorphic human RMS not available for the analysis? Based on the comparison of NPE and ARMS/ERMS signatures, the authors then frequently state that NPE share a more ARMS-related phenotype, which is very puzzling for the reader. As far as I understood, the reasoning of the authors is as follows: Since ARMS are more aggressive than ERMS, the data support the observation that Neu1 haploinsufficiency promotes the development of more poorly differentiated phenotype, which fits their histologically pleomorphic appearance. The

authors must discuss this line of thought in much more detail in the discussion section.

- Nevertheless, I do not agree with the interpretation of the authors regarding location of the tumors and the data shown in Fig 2h. According to the heat map shown in Fig2h – and as stated in line 158 of the manuscript – the highest correlation is between PE/NPE and ERMS. There are only 9 genes that (partly) correlate with fusion-positive ARMS. The suggestion that PE and NPE tumors are ARMS-related is also based on the observation that these tumors develop in either the extremities or the trunk of the mice. The authors state that this reflects the location of human ARMS with poor prognosis and short survival. However, please note that tumors of Ptch1+/- mice most frequently occur at the extremities and the trunk although they histologically and molecularly resemble human ERMS (e.g. see Nitzki et al, *Oncogene* 2016, 35, 2923–2931; Rubin et al, *Cancer Cell* 19, 177–191 February 15, 2011; Kappler et al, *Oncogene* 2004, 23, 8785–8795). Thus, the expression of the 9 ARMS-related genes and the location of the tumors may just as well reflect species-specific differences. This needs discussion. In addition, it is also puzzling that PE and NPE tumors are not different in the analysis shown in Fig2h, whereas they are highly different in multiplex flow cytometry and microarray analysis of individual NPE and PE limb and trunk RMS. This also needs a statement.

- I also would change the following headings and replace them as follows:

o The title “Neu1+/-/Ptch1+/-/ ETV7 TG+/- mice are a model of human RMS” does not tell anything. I would write “Neu1+/-/Ptch1+/-/ ETV7 TG+/- mice share molecular characteristics of human ARMS and ERMS”

o The title “Human and mouse RMS share an adipogenic signature” is also not distinct. Better is something like “Pleomorphic Neu1+/-/Ptch1+/-/ETV7+/-/TG RMS share an adipogenic signature with human ARMS and ERMS”.

- In Line 102 the authors write:.... was sufficient to increase the incidence of RMS to 62%, as compared to 54% in the.... Please indicate if this difference is significant.

- Lines 106-109:....Within the cohort of mice with RMS in the ribcage, 4/35 (11.4%) NPE and 1/23 (4.3%) PE mice also developed secondary growths in the esophagus, peritoneum, forelimb, and flank (Fig 1a, c), a finding indicative of metastatic spread.... Why is this indicative for metastatic spread? Single tumors in Ptch+/- mice are also found at these locations (see above) and could be as well synchronous or metachronous tumors.

- Line 193: total LAMP1 intensity shown in Fig.3g is not convincing. Please analyze more than 4 tumors as you did in all other assays.

- Line 206: ... Protein expression data were generated by using Lamp1PM in combination with markers selective for each cell population... Please provide literature for the markers used in the study.

- Line 209: To uncover distinct cell populations within each tumor sample, we narrowed the tSNE analysis to tumor and stromal cells by excluding hematopoietic cells using CD45.... I do not understand why then F4/80 and B220 were used in the multiplex analysis. Isn't it like that CD45+ cells also comprise B220+ and F4/80+ cells?

- Line 213:....These analyses showed that the cell clusters differed not only between NPE and PE RMS, independently of their CD45 status, but also between limb and trunk tumors (Fig. 4c)....Please provide an explanation for this finding.

- Line 220:These cells were further analyzed by using Lamp1PM expression in combination with a set of canonical markers for cancer and stromal cells (Fig. 5 c-h)....The only markers, which distinguish between cancer and stromal cells are Lyve1 and EpCAM. Is this enough?

- Line 223:Within the annotated populations, the total percentage of Lamp1PMpos, exocytic cells was 5.3% in limb and 3.0% in trunk RMS (Fig. 5e-h)....I do not really understand: why 5.3% and 3.0%? In Fig. 5g the numbers of exocytic cells seem to be much higher.

- Line 230:These exocytic, pleomorphic subpopulations accounted for 21% of cancer cells in the limb and 15% in the trunk (populations 1 - 6) and were more abundant in NPE than PE RMS (Fig. 5e-h)....What does this suggest?

- Line 233:Similarly, within the stromal CD45neg and Desneg cell populations, the majority of Lamp1PMpos CAFs (cancer associated fibroblasts) also expressed markers typical of....How do you know, which cells are CAFs and which are not?

- Line 246:.... suggest that in NPE tumors not only cancer cells but also stromal cells are maintained in an intermediate state of transdifferentiation that could be at the basis of their

distinct pleomorphic phenotype.....what about normal muscle of NPE mice? Since Neu1^{-/-} mice have a severe phenotype regarding ECM and connective tissue, normal muscle tissue of Neu1^{+/-} or NPE mice must be analyzed to make sure that the described phenotype is indeed related to the tumors.

- Line 259:upregulated genes in NPE RMS belong to pathways of lipid metabolism, myogenesis and EMT (epithelial-to-mesenchymal transition)....needs to be analyzed in normal muscle tissue of the mice as well.

- Line 283:... The earliest progenitor marker of myogenesis, Pax3... please provide a reference

- Line 312: ...Based on these data, we revisited the H&E and Masson's trichrome stained NPE RMS and identified increased numbers of adipose cells....Please provide the numbers by counting these cells in PE and NPE tumors.

- Line 345: ... We have successfully developed the first, spontaneously occurring model of pleomorphic RMS in mice by lowering the expression levels of the lysosomal sialidase Neu1 in the Ptch1^{+/-}/ETV7Tg^{+/-} genetic background.....What does "spontaneously" mean? Does it mean without the use of conditional alleles? This is important because there are other models for pleomorphic RMS (e.g. see "Cooperation of oncogenic K-ras and p53 deficiency in pleomorphic rhabdomyosarcoma development in adult mice" published 2006 by Tsumura H and colleagues).

- Line 374:Furthermore, the overlooked adipose component in these tumors underscores the importance of combining the use of adipose markers during diagnosis of this type of RMS....please discuss these data in light of the paper "A mouse model of rhabdomyosarcoma originating from the adipocyte lineage" published by Mark E. Hatley and colleagues.

- Line 382: ... Neu1^{+/-} FVB/NJ mice...please provide a reference

Minor:

Figure legends: it would be easier for the reader if a)b)c) etc are consistently either in front or at the end of a sentence

Fig. 1: please omit **** P ≤ 0.0001

Fig. 4: "c)" is missing

Reviewer #2 (Remarks to the Author):

Review of Machado et al, "Modelling pleomorphic rhabdomyosarcoma in mice by haploinsufficiency of the lysosomal sialidase NEU1"

Summary

This work builds off of previous work from the corresponding authors, either studying lysosomal exocytosis in a Arf^{-/-};Neu^{+/-} deficient mouse models of pleomorphic sarcomas, or ETV7 and its role in rapamycin insensitive mTOR complexes in many cancers, including rhabdomyosarcoma (RMS). This collaborative effort implements a new combination of alleles and genetic mouse modeling of pleomorphic rhabdomyosarcoma, a rare RMS sub-type that predominantly occurs in adults, is difficult to treat, and has poor overall survival. This was an exciting new model that had a high incidence of pleomorphic RMS, and generated mechanistic data to detail how NEU1 deficiency skewed developing tumors towards a RMS phenotype. Previous pleomorphic RMS models have relied on oncogenic K-ras and p53 deficiency to generate pleomorphic rhabdomyosarcoma, thus this model introduces new cooperating genetic events that generate this rare tumor.

Specifically, the authors implemented a previous mouse model that had a low incidence of RMS because of Ptch1 deficiency. They added overexpression of human ETV7 as well as NEU1 deficiency. These allelic combinations are referred to as PE (Patch^{+/-}; ETV7Tg^{+/-}) or NPE (Patch^{+/-}; ETV7Tg^{+/-};Neu^{+/-}). NEU1 is a lysosomal sialidase that is suppressed in many cancer types, and results in a degradation of plasma membrane integrity and the extracellular matrix. Here, the authors show that Neu1 deficiency results in an enhanced incidence of pleomorphic RMS, and that NPE tumors have increased fibrosis and collagen staining, and a redistribution of Lamp1 at the plasma membrane of tumor and stromal cells. They perform single cell flow analysis to determine that the tumors are both NP and NPE tumor types are very heterogenous and that Lamp1 plasma membrane positive cells are enriched in exocytic and undifferentiated NPE RMS.

Further, they determine that NPE has an adipogenic gene signature associated with less differentiated characteristics that is not present in NP, and suggest that this is a shared feature, especially with the alveolar RMS subtype.

With regards to comparison to the human disease, the authors use a Human 2.0 ST Microarray on their mouse tumor samples, and compare to data from pediatric RMS sub-types, embryonal and alveolar. I recognize that this is because pleomorphic RMS expression data may be unavailable. They show that: 1) NPE vs PE exhibit few gene expression differences, and 2) there are shared features with both ERMS and ARMS. Further, NPE and PE both express the classical clinical markers, desmin, myod, and myogenin, and there is a pleomorphic histology. However, more gene expression analysis should be performed (with pre-existing data) to put this model in the context of other mouse rhabdomyosarcoma models, and enhance the argument for a separate, pleomorphic subtype.

Overall, this is an interesting new mouse model and mechanism to understand the pleomorphic sarcoma sub-type, and to compare how its molecular features are different or shared across other RMS sub-types. This study will be of interest to the field.

Specific Comments:

1) Contextual comparison of pleomorphic RMS to other RMS mouse models. Pleomorphic RMS presents in adult patients, whereas ARMS and ERMS are in pediatric populations. How does the presentation of this model differ from other RMS models? Is the latency similar? Parallels are made with ARMS, and understanding how this model differs from other pediatric sarcoma mouse models could bolster the argument for shared adipogenic features in pleomorphic RMS and ARMS vs. ERMS. To address this, a differential gene expression analysis/PCA/hierarchical clustering of NPE and PE microarray data with previously existing mouse RMS microarray data would support the rationale for this representing pleomorphic RMS/a distinct entity (microarray data from other RMS mouse models is available from Geo Database: GSE22520).

2) Criteria for defining a pleomorphic RMS vs other sub-type diagnosis in the mouse model. Figure 1a details all of the cases for RMS, but it's not clear how these are being defined. I'm assuming this is based off of H&E staining and pathological review, as well as MyoD, MyoG, and Desmin reactivity, but this is not explicitly stated in the text and is an important basis for the paper. Re-write for clarity. It would also be interesting to know what the other tumor types were that developed in PE and NPE if this data is available. Were the other tumors types (outside of RMS) similar between NPE and PE, or different?

3) Neu1 deficiency contributes to more aggressive disease in other sarcoma sub-types, but in the PE vs NPE mouse model there is identical survival. Although the NPE tumors have more aggressive features, why do you think this survival is identical? Could the effect of Neu1 be most relevant when mitigating efficacy of cytotoxic chemotherapy agents used for treatment rather than for disease onset. This is outside the scope of this manuscript to address experimentally but is an interesting idea and this discrepancy could be commented on in the discussion.

Minor Comments:

1) Figure 2C – this is presented as a representative image for an increase in connective tissue in RMS TMAs, but it's not clear what the comparison is to. Also, the composition of RMS tumors on the TMA is not clear (how many ARMS, ERMS, ssRMS?), or how many of the analyzed tumors had this increase in connective tissue?

2) Is the Figure 5E-F heatmap for high/low flipped for red/blue? I didn't see the lines 226-229 being true if not flipped, or it wasn't clear. Mainly the demarcation from Lamp1 being positive or negative is opposite of what I would think based on the clusters.

3) Referencing specific clusters for Figure 5 within the text would help reader understanding

4) Figure 6- the focus of this figure is on overexpressed genes, but were any patterns seen with the NPE downregulated genes?

- 5) In the methods section it says the Human 2.0 ST array was used for mouse tumor samples— what was the rationale? How many of the human microarray probes align to the mouse genome?
- 6) Line 134 – what myogenic markers are being referred to for determining “differentiation status” in this context.
- 7) Extended data is not referenced in order.

Manuscript #. COMMSBIO-21-2625A

Reply to Reviewers' comments

We thank both Reviewers for their careful reading of our manuscript and their insightful suggestions and comments.

Reviewer #1 (Remarks to the Author):

The authors of the manuscript "Modelling pleomorphic rhabdomyosarcoma in mice by haploinsufficiency of the 2 lysosomal sialidase NEU1" studied the effect of Neu1 haploinsufficiency in *Ptch1^{+/-}/ETV7TG^{+/-}* (PE) mice that develop rhabdomyosarcoma (RMS) with high incidence. Neu1 is a lysosomal sialidase that negatively regulates exocytosis of lysosomes by cleaving the sialic acids of LAMP1. Thus, low NEU1 activity increases the number of lysosomes containing sialylated LAMP1 that dock at the plasma membrane. This finally results in excessive lysosomal exocytosis and leads to disruption of the integrity of the plasma membrane and extra cellular matrix.

The authors show that RMS in PE mice are different from those in *Neu1^{+/-} Ptch1^{+/-}/ETV7TG^{+/-}* (NPE) mice. Thus, whereas PE tumors are more differentiated, the RMS of NPE mice show cellular heterogeneity and numerous undifferentiated, anaplastic foci and are rich in pleomorphic and rhabdoid cells. In addition, connective tissue and intracytoplasmic collagen are elevated in NPE tumors suggesting that reduced Neu1 expression promotes pleomorphism and transforms the tumor stroma into a desmoplastic/fibrotic state.

The authors also show that low expression of NEU1 is accompanied by an overall increase in LAMP1 in murine (and also human) RMS. In addition, examination of b-hexosaminidase in the interstitial fluid of several NPE RMS confirmed that NPE tumors are more exocytic than PE RMS. Next, the authors performed multiplex flow cytometry. The data show that the tumors are composed of discrete cell clusters that differ not only between NPE and PE RMS, but also between limb and trunk tumors. Moreover, protein expression in selected CD45^{neg} LAMP1^{pos} cell populations suggest that cancer and stromal cells in NPE tumors are maintained in an intermediate state of transdifferentiation. Finally, GSEA analysis suggested that this intermediate differentiation and pleomorphism of NPE RMS have a strong adipogenic component.

The paper is very well written. All methods are described in detail and statistical analyses are adequate.

Although the study is predominantly descriptive, it contains interesting data.

However, some points need to be addressed:

1. Inverse expression of Neu1 and LAMP1 has already been described in sarcoma including RMS. Similarly, it has been demonstrated that Neu1 haploinsufficiency fosters the development

of pleomorphic sarcomas and that human RMS cells with low Neu1 and high LAMP1 show excessive lysosomal exocytosis (Machado et al. *Sci. Adv.* 2015;1:e1500603 18 December 2015). Therefore, the results shown in the current paper are not really new. The authors should comment on that.

R. The experimental set up of the current study is different, as the mice mentioned in our *Science Adv.* paper were *Arf^{-/-} Neu1^{+/-}*, which developed a broad range of sarcoma and hematologic malignancies, while the mice used here were on a *Ptch1^{+/-}* background, which predisposes to RMS. We have chosen this experimental model to specifically address the additive effects of Neu1 haploinsufficiency over the *Ptch1^{+/-}/ETV7* (PE) background. The high frequency of RMS, occurring in the PE mice at early age made the comparative analyses with NPE mice more accurate and cleaner. RMS does express ETV7 (Harwood, F. C. et al. *ETV7 is an essential component of a rapamycin-insensitive mTOR complex in cancer. Sci Adv 4, eaar3938, doi:10.1126/sciadv.aar3938 (2018)*).

2. Neu1^{-/-} mice show enhanced infiltration with connective tissue and alterations of the ECM (Zanoteli et al; *Biochim Biophys Acta.* 2010 ; 1802(7-8): 659–672. doi:10.1016/j.bbadis.2010.04.002.). Therefore, I am wondering whether normal muscle of Neu1^{+/-} mice also shows these characteristics. If so these characteristics are not a RMS phenotype, but rather are related to the Neu1 mutation itself. Therefore, it is important to repeat the experiments shown in Fig.2a – Fig.2g on normal muscle tissue of NPE mice.

R. In reply to this pertinent comment by the Reviewer, we have performed histological Masson's Trichrome staining (Extended Data Fig. 2c) and tested the gene expression of *Col1a2* and *Col4a1* (Extended Data Fig. 2d and e) in PE and NPE normal muscle. We found neither expansion of the connective tissue nor altered gene expression in any of the PE and NPE normal tissues.

Therefore, the effects of *Neu1* haploinsufficiency are restricted to the tumors, where the levels of the enzyme appeared to be further downregulated, a phenomenon that is consistent with that observed in other human tumor types (Machado et al 2015). In this context, I need to clarify that *Neu1* heterozygosity in mice, as it is the case in human carriers, does not result in a sialidosis phenotype.

We have added these data to the Results section of the revised manuscript (Line 171-174): “These combined observations were not seen in normal muscle tissue of these mice (Extended data Fig. 2c-e), emphasizing that the connective tissue deposition observed in the NPE RMS tumors was caused by the combination of reduced activity of Neu1 within tumor cells and stroma.”

Extended Data Fig. 2 | Connective tissue expression in normal PE and NPE muscles. a and b, Quantification of ECM/collagen deposition (a) and intracytoplasmic (b) in ARMS, ERMS and spindle cell RMS TMA sections. **c,** Masson's Trichrome staining of normal PE and NPE muscles. **d and e,** *Col1a2* mRNA expression in normal PE and NPE skeletal muscle. Mean \pm s.d.; Welch (unpaired) *t*-test; $n=12$ (PE) and $n=14$ (NPE). **g,** *Col4a1* mRNA expression in normal PE and NPE skeletal muscle. Mean \pm s.d.; Welch (unpaired) *t*-test; $n=11$ (PE) and $n=12$ (NPE).

3. Please provide an information about the genetic background of the mice and indicate whether the PE and NPE mice are littermates. This is a very important information, because the genetic background may influence some of the phenotypic characteristics of the tumors.

R. The mice are on a mixed background FVB/129sv/C57BL/6. The study has been performed over 2 years and both PE and NPE mice were obtained from the same strategic breeding schedules (*Neu1^{+/-}/Ptch1^{+/-}/ETV7^{TG+/-}* X *Neu1^{+/-}/Ptch1^{+/-}/ETV7^{TG+/+}*), but a NPE mouse with a tumor not necessarily came from the same litter as a PE mouse with a tumor. We agree with the reviewer that the genetic background may influence some of the phenotypic characteristics. Likely, none of the RMS tumors have the exact same genetic background, as RMS patients have dissimilar genetic backgrounds. Despite the slight genetic heterogeneity, the phenotypic outcome is similar in our cohort of mice.

We have added a more detailed description of the mice to the Method section of the manuscript as follows (Line 495-500): "*Neu1^{+/-}* FVB/NJ mice were crossed with *Ptch1^{+/-}/ETV7^{TG+/-}* (129sv/C57BL/6) mice. The mice used in this study were obtained (over a timespan of 2 years) from the same strategic breeding schedule: *Neu1^{+/-}/Ptch1^{+/-}/ETV7^{TG+/-}* X *Neu1^{+/-}/Ptch1^{+/-}/ETV7^{TG+/+}*, but PE and NPE mice with tumors not necessarily came from the same litter.

4. Based on histology I agree with the authors that NPE tumors resemble human pleomorphic RMS. Therefore, I am wondering why the authors compared the gene/protein expression signatures of NPE tumors to that of human ARMS and ERMS but not to the signature of pleomorphic RMS. One example is the adipocytic signature. Were pleomorphic human RMS not

available for the analysis? Based on the comparison of NPE and ARMS/ERMS signatures, the authors then frequently state that NPE share a more ARMS-related phenotype, which is very puzzling for the reader. As far as I understood, the reasoning of the authors is as follows: Since ARMS are more aggressive than ERMS, the data support the observation that Neu1 haploinsufficiency promotes the development of more poorly differentiated phenotype, which fits their histologically pleomorphic appearance. The authors must discuss this line of thought in much more detail in the discussion section.

R. Indeed the Reviewer is correct in that we could not align our gene expression data sets to those of human pleomorphic RMS, because there are no expression data sets available for this subtype. We also concur with the Reviewer that our mouse expression data align better with those of ERMS than ARMS, which showed only 10% overlap (9 out of 91 genes). We have rephrased the text to make this point clearer as follows:

Results Line 193-199 *“Within this gene set, the highest positive correlation was found between the expression profiles of NPE ($r = 0.71$) and PE ($r = 0.72$) tumors with human ERMS that comprise the most genetically heterogeneous subtypes⁴ (Fig. 2h). Although no significant differences at the transcriptional levels were observed between NPE or PE tumors compared to human RMS, our data support the notion that Neu1 haploinsufficiency in the NPE model promotes the development of a more poorly differentiated phenotype, as demonstrated by their histologically pleomorphic appearance.”*

Discussion Line 451-456 *“Although NPE tumors arise preferentially in the limbs and trunk, and only express 10% of the genes that trended with human ARMS, their genetic features are much closer to ERMS. Unfortunately, the unavailability of compatible genetic data from human pleomorphic RMS that we could compare to our NPE data sets, limits our current evaluation of NPE tumors to their histological characteristics and further investigation is required to confirm NPE mice as model of human pleomorphic RMS”.*

5a. Nevertheless, I do not agree with the interpretation of the authors regarding location of the tumors and the data shown in Fig 2h. According to the heat map shown in Fig2h – and as stated in line 158 of the manuscript - the highest correlation is between PE/NPE and ERMS. There are only 9 genes that (partly) correlate with fusion-positive ARMS. The suggestion that PE and NPE tumors are ARMS-related is also based on the observation that these tumors develop in either the extremities or the trunk of the mice. The authors state that this reflects the location of human ARMS with poor prognosis and short survival. However, please note that tumors of Ptch1^{+/-} mice most frequently occur at the extremities and the trunk although they histologically and molecularly resemble human ERMS (e.g. see Nitzki et al, Oncogene 2016, 35, 2923–2931; Rubin et al, Cancer Cell 19, 177–191 February 15, 2011; Kappler et al, Oncogene 2004, 23, 8785–8795). Thus, the expression of the 9 ARMS-related genes and the location of the tumors may just as well reflect species-specific differences. This needs discussion.

R. The reviewer is correct, we have changed the text to emphasize that the highest correlation between our murine models is with human ERMS and that there were no significant differences in gene expression profiles comparing PE or NPE tumors with human RMS.

We have corrected the text as follows:

Results; Line 189-196: *“As shown in the heatmap (Fig. 2h), the pool of human genes that could be analyzed and compared were 91 genes from different platforms (murine microarray and Pediatric Cancer Genome Project (PCGP) RNAseq) that passed the principal component analysis (PCA), quality control, and significant Pearson correlation criteria. This comparative analysis showed no significant differences between human RMS and our NPE and PE tumors. Within this gene set, the highest positive correlation was found between the expression profiles of NPE ($r = 0.71$) and PE ($r = 0.72$) tumors with human ERMS that comprise the most genetically heterogeneous subtypes⁴ (Fig. 2h).”*

5b. In addition, it is also puzzling that PE and NPE tumors are not different in the analysis shown in Fig2h, whereas they are highly different in multiplex flow cytometry and microarray analysis of individual NPE and PE limb and trunk RMS. This also needs a statement.

R. The reviewer is correct. The heatmap shown in Fig. 2h is a comparison of gene expression between our murine models and human RMS. Instead, the protein markers used for flow cytometry analyses to compare PE and NPE tumors were chosen to specifically discriminate between different populations or clusters of cells and were based on differential expression of posttranslationally modified cell surface antigens rather than gene transcription. We have attributed the differences observed between PE and NPE analyzed via microchip arrays to changes in tumor phenotype as a consequence of *Neu1* haploinsufficiency.

We have added the following statement in the Results section (Line 189-193): *“As shown in the heatmap (Fig. 2h), the pool of human genes that could be analyzed and compared were 91 genes from different platforms (murine microarray and Pediatric Cancer Genome Project (PCGP) RNAseq) that passed the principal component analysis (PCA), quality control, and significant Pearson correlation criteria. This comparative analysis showed no significant differences between human RMS and our NPE and PE tumors.”*

6. I also would change the following headings and replace them as follows:

6a. The title “*Neu1*^{+/-}/*Ptch1*^{+/-}/ *ETV7* TG^{+/-} mice are a model of human RMS” does not tell anything. I would write “*Neu1*^{+/-}/*Ptch1*^{+/-}/ *ETV7* TG^{+/-} mice share molecular characteristics of human ARMS and ERMS”

R. Following the reviewers' suggestion we have now changed the subtitle to: “*Neu1*^{+/-}/*Ptch1*^{+/-}/*ETV7*^{TG+/-} mice share molecular characteristics of human ARMS and ERMS” (Line 185)

6b. The title “Human and mouse RMS share an adipogenic signature” is also not distinct. Better is something like “Pleomorphic *Neu1*^{+/-}/*Ptch1*^{+/-}/*ETV7*^{+/-}/TG RMS share an adipogenic signature with human ARMS and ERMS”.

R. Following the reviewers' suggestion, we have now changed the subtitle to (Line 403): “Pleomorphic *Neu1*^{+/-}/*Ptch1*^{+/-}/*ETV7*^{TG+/-} RMS share an adipogenic signature with human ARMS and ERMS”.

7. In Line 102 the authors write:.... was sufficient to increase the incidence of RMS to 62%, as compared to 54% in the.... Please indicate if this difference is significant.

R. Our bioinformatics person (SMD) responded that this represents the total number (%) of RMS tumors in NPE and PE mice and therefore statistical analysis cannot be performed.

8. Lines 106-109:....Within the cohort of mice with RMS in the ribcage, 4/35 (11.4%) NPE and 1/23 (4.3%) PE mice also developed secondary growths in the esophagus, peritoneum, forelimb, and flank (Fig 1a, c), a finding indicative of metastatic spread... Why is this indicative for metastatic spread? Single tumors in *Ptch*^{+/-} mice are also found at these locations (see above) and could be as well synchronous or metachronous tumors.

R. The reviewer has a valid point and we have rephrased the text as follows (Line 110-113):

“Within the cohort of mice with RMS tumors, 4/35 (11.4%) NPE and 1/23 (4.3%) PE mice also developed secondary growths (Fig 1a, c), a finding that may indicate metastatic spread, although without additional genomic analysis we cannot exclude that these tumors are synchronous rather than metachronous.”

9. Line 193: total LAMP1 intensity shown in Fig.3g is not convincing. Please analyze more than 4 tumors as you did in all other assays.

R. We have now analyzed additional samples, 5x *Neu1*^{+/+}/*Ptch1*^{+/-}/*ETV7*^{TG+/-} and 5x *Neu1*^{+/-}/*Ptch1*^{+/-}/*ETV7*^{TG+/-} and changed Figure 3g accordingly.

Fig. 3 | LAMP1 redistribution at the PM is a readout for increased lysosomal exocytosis downstream of low NEU1 expression. **a**, Representative core of human RMS TMAs immunostained for LAMP1. **b**, Density map of algorithm applied to a LAMP1-stained core (a) showing LAMP1^{PM} minimal and maximum density and location. **c**, Heat-map representing the intensity of LAMP1^{PM} staining of the core shown in a. **d**, Correlation between low NEU1 and high LAMP1^{PM} immunoreactivity in paired cores of human RMS TMAs. TMA (2) were used; Paired *t*-test; *n*=41. **e**, Representative micrographs of increased Lamp1 immunostaining in NPE compared with PE RMS. Punctuated lysosomal staining, typical for Lamp1. **f**, Flow cytometry histogram showing higher Lamp1^{PMpos} expression in cells in NPE RMS versus PE RMS. **g**, Increased Lamp1^{PM} intensity quantified by flow cytometry in NPE versus PE RMS. Paired *t*-test; *n*=4. **h**, Increased b-hexosaminidase activity in the interstitial fluid from NPE versus PE RMS. Student's (unpaired) *t*-test; *n*=7.

10. Line 206: ... Protein expression data were generated by using Lamp1PM in combination with markers selective for each cell population... Please provide literature for the markers used in the study.

R. Following the reviewers' suggestion we have added references to the manuscript for the markers used in this study (Line 263).

Park, J. W. *et al.* Stem Cells Antigen-1 Enriches for a Cancer Stem Cell-Like Subpopulation in Mouse Gastric Cancer. *Stem Cells*. 2016 May;34(5):1177-87. doi: 10.1002/stem.2329.

Pérot, G. *et al.* Smooth muscle differentiation identifies two classes of poorly differentiated pleomorphic sarcomas with distinct outcome. *Mod Pathol*. 2014 Jun;27(6):840-50. doi: 10.1038/modpathol.2013.205. Epub 2013 Nov 29.

Furlong, M.A. *et al.* Pleomorphic Rhabdomyosarcoma in Adults: A Clinicopathologic Study of 38 Cases with Emphasis on Morphologic Variants and Recent Skeletal Muscle- Specific Markers. *Mod Pathol*. 2001 Jun;14(6):595-603. doi: 10.1038/modpathol.3880357.

Lee, E. *et al.* Crosstalk between cancer cells and blood endothelial and lymphatic endothelial cells in tumour and organ microenvironment. *Expert Rev Mol Med*. 2015 Jan 30;17:e3. doi: 10.1017/erm.2015.2.

Sahin, A.A. *et al.* Tumor proliferative fraction in solid malignant neoplasms. A comparative study of Ki-67 immunostaining and flow cytometric determinations. *Am J Clin Pathol*. 1991 Oct;96(4):512-9. doi: 10.1093/ajcp/96.4.512.

Hyun, K-A. *et al.* Epithelial-to-mesenchymal transition leads to loss of EpCAM and different physical properties in circulating tumor cells from metastatic breast cancer. *Oncotarget*. 2016 Apr 26;7(17):24677-87. doi: 10.18632/oncotarget.8250.

Truong, L.D. *et al.* The Diagnostic Utility of Desmin. A study of 584 cases and review of the literature. *Am J Clin Pathol*. 1990 Mar;93(3):305-14. doi: 10.1093/ajcp/93.3.305.

Chen, C. *et al.* The biology and role of CD44 in cancer progression: therapeutic implications. *J Hematol Oncol*. 2018 May 10;11(1):64. doi: 10.1186/s13045-018-0605-5

Sapino, A. *et al.* Expression of CD31 by cells of extensive ductal in situ and invasive carcinomas of the breast. *J Pathol*. 2001 Jun;194(2):254-61. doi: 10.1002/1096-9896(200106)194:2<254::AID-PATH880>3.0.CO;2-2.

Zhang, Y-Y. *et al.* CD31 regulates metastasis by inducing epithelial–mesenchymal transition in hepatocellular carcinoma via the ITGB1-FAK-Akt signaling pathway. *Cancer Lett*. 2018 Aug 10;429:29-40. doi: 10.1016/j.canlet.2018.05.004. Epub 2018 May 8.

Han, C. *et al.* Biomarkers for cancer-associated fibroblasts. *Biomarker Research* (2020)8:64

Misharin, A.V. *et al.* Flow Cytometric Analysis of Macrophages and Dendritic Cell Subsets in the Mouse Lung. *Am J Respir Cell Mol Biol*. 2013 Oct; 49(4): 503–510. doi: 10.1165/rcmb.2013-0086MA

Mukai, K. et al. Critical role of P1-Runx1 in mouse basophil development. *Blood*. 2012 Jul 5; 120(1): 76–85. doi: 10.1182/blood-2011-12-399113

Yui, M.A. and Rothenberg, E.V. Developmental gene networks: a triathlon on the course to T cell identity. *Nat Rev Immunol*. 2014 Aug; 14(8): 529–545. doi: 10.1038/nri3702

11. Line 209: To uncover distinct cell populations within each tumor sample, we narrowed the tSNE analysis to tumor and stromal cells by excluding hematopoietic cells using CD45.... I do not understand why then F4/80 and B220 were used in the multiplex analysis. Isn't it like that CD45+ cells also comprise B220+ and F4/80+ cells?

R. For the tSNE analysis we have used the CD45 marker to exclude all hematopoietic cells from the tumor and stromal cell populations. In addition, the multiplex analysis was used to separately determine the percentages of macrophages, T cells and B cells in CD45^{pos} cells (Extended Data Fig. 6).

12. Line 213:....These analyses showed that the cell clusters differed not only between NPE and PE RMS, independently of their CD45 status, but also between limb and trunk tumors (Fig. 4c)....Please provide an explanation for this finding.

R. Although both limb and trunk tumors were bona fide RMS by histopathology, they nonetheless can still have a different marker signature. The point we wanted to make was that the differences seen between limb and trunk tumors in mice might reflect similar differences in the corresponding human tumors. In addition, the tumor microenvironment and cell of origin of the tumors might influence the differences seen between the two locations.

We have added a statement to the text to better explain these findings (Line 268-270).

"This may be linked to the cell of origin that supports the development of trunk versus limb tumors, but a more comprehensive investigation would be necessary to confirm these differences."

13. Line 220:These cells were further analyzed by using Lamp1PM expression in combination with a set of canonical markers for cancer and stromal cells (Fig. 5 c-h)....The only markers, which distinguish between cancer and stromal cells are Lyve1 and EpCAM. Is this enough?

R. We did not rely on individual markers to make the distinction between cancer and stromal cells, but rather on a combined set of markers that were absent or present (i.e. desmin, Ki67, Sca1, Sma, Pcam1, CD44, Lyve1 and EpCam).

14. Line 223:Within the annotated populations, the total percentage of Lamp1PM^{pos}, exocytic cells was 5.3% in limb and 3.0% in trunk RMS (Fig. 5e-h)....I do not really understand: why 5.3% and 3.0%? In Fig. 5g the numbers of exocytic cells seem to be much higher.

R. The reviewer is correct, and we have now changed the text with the correct percentages for the limb (45.95%) and trunk (11.14%). Line 281-281

15. Line 230:These exocytic, pleomorphic subpopulations accounted for 21% of cancer cells in the limb and 15% in the trunk (populations 1 - 6) and were more abundant in NPE than PE RMS (Fig. 5e-h)....What does this suggest?

R. The reviewer highlights an interesting point. We have currently no clear explanation for this finding, but we can hypothesize that NPE tumors are poorly differentiated and more proliferative, have increased *Myc* expression and therefore suppress expression of lysosomal genes including *Neu1* (Annunziata, I. et al. *MYC competes with Mit/TFE in regulating lysosomal biogenesis and autophagy through an epigenetic rheostat. Nature Communications. 2019 Aug 9;10(1):3623. doi: 10.1038/s41467-019-11568-0*).

We have added a conclusion to the text (Line 289-292): *“The combination of pleomorphic and exocytic cells enriched in NPE tumors indicates that Neu1 haploinsufficiency promotes the poorly differentiated state of the tumors, albeit an in depth explanation of this phenomenon may require further investigation.”*

16. Line 233:Similarly, within the stromal CD45neg and Desneg cell populations, the majority of Lamp1PMpos CAFs (cancer associated fibroblasts) also expressed markers typical of....How do you know, which cells are CAFs and which are not?

R. As these cells are residing in the tumor and are highly positive for SMA we assumed that they are cancer associated fibroblasts. We have changed the text to better explain this point Line 293-297.

“Similarly, within the stromal CD45^{neg} and Des^{neg} cell populations, the majority of Lamp1^{PMpos} cells included CAFs (SMA) (cancer associated fibroblasts) (cell cluster 7–9) and epithelial cells (cell clusters 10–11 in limb and 10 in trunk), but also expressed markers typical of myofibroblasts/mesenchymal cells, e.g., Ki67, Sca1, Sma, Pcam1, CD44, Lyve1 and EpCam (epithelial) (Fig. 5e-h).”

17. Line 246:.... suggest that in NPE tumors not only cancer cells but also stromal cells are maintained in an intermediate state of transdifferentiation that could be at the basis of their distinct pleomorphic phenotype.....what about normal muscle of NPE mice? Since Neu1-/- mice have a severe phenotype regarding ECM and connective tissue, normal muscle tissue of Neu1+/- or NPE mice must be analyzed to make sure that the described phenotype is indeed related to the tumors.

R. As mentioned in our response to point 2, we have stained normal PE and NPE skeletal muscle sections with Masson's Trichrome and ran RT-qPCR for *Col1a2* and *Col4a1*. We have not observed the accumulation of connective tissue in any of the normal tissues of PE and NPE mice (Extended Data Fig. 2c-e).

Line 171-174: *“These combined observations were not seen in normal muscle tissue of these mice (Extended data Fig. 2c-e), emphasizing that the connective tissue deposition observed in the NPE RMS tumors was caused by the combination of reduced activity of Neu1 within tumor cells and stroma.”*

Also, as stated earlier, *Neu1* heterozygosity in mice, as it is the case in human carriers, does not result in a sialidosis phenotype.

Extended Data Fig. 2 | Connective tissue expression in normal PE and NPE muscles. a and b, Quantification of ECM/collagen deposition (a) and intracytoplasmic (b) in ARMS, ERMS and spindle cell RMS TMA sections. **c,** Masson's Trichrome staining of normal PE and NPE muscles. **d and e,** *Col1a2* mRNA expression in normal PE and NPE skeletal muscle. Mean \pm s.d.; Welch (unpaired) *t*-test; $n=12$ (PE) and $n=14$ (NPE). **g,** *Col4a1* mRNA expression in normal PE and NPE skeletal muscle. Mean \pm s.d.; Welch (unpaired) *t*-test; $n=11$ (PE) and $n=12$ (NPE).

18. Line 259:upregulated genes in NPE RMS belong to pathways of lipid metabolism, myogenesis and EMT (epithelial-to-mesenchymal transition)...needs to be analyzed in normal muscle tissue of the mice as well.

R. Per reviewers' suggestion we have now included RT-qPCR analyses for all genes defining adipogenesis, myogenesis and EMT and found no differences in expression between normal PE and NPE muscle samples (Extended Data Fig. 8). The text was adjusted accordingly (Line 397-400).

"To exclude the possibility that *Neu1* haploinsufficiency by itself can cause the significant changes in gene expression observed between PE and NPE tumors, we performed RT-qPCR analysis of the same set of genes in normal muscle tissue from both NPE and PE mice and found no differences (Extended Data Fig. 8)."

Extended Data Fig. 8 | Gene expression in normal PE and NPE skeletal muscle. a–c, qRT-PCR analysis of normal PE and NPE skeletal muscle for genes involved in adipogenesis (a) myogenesis (b) and tumor aggressiveness (c). qRT-PCR results are from normalized mRNA expression relative to normal PE muscle. Mean \pm s.d.; Welch (unpaired) *t*-test; $n \geq 6$.

19. Line 283:… The earliest progenitor marker of myogenesis, Pax3… please provide a reference

R. We have added the reference: “*Origin of Vertebrate Limb Muscle: The Role of Progenitor and Myoblast Populations (Chapter one); Malea Murphy and Gabrielle Kardon 2011; Current Topics in Development Biology*” to the text (line 369)

20. Line 312: …Based on these data, we revisited the H&E and Masson’s trichrome stained NPE RMS and identified increased numbers of adipose cells….Please provide the numbers by counting these cells in PE and NPE tumors.

R. We have adjusted the text in line 406-410 to: “*Based on these data, we revisited the H&E and Masson’s trichrome stained PE and NPE RMS and identified a slight increase in the percentage of adipose cells in the NPE sections, although this difference was not statistically significant (Extended Data Fig. 9a).*”

Extended Data Fig. 9 | GSEA between RMS patients with and Hallmark and KEGG pathways. a, Quantification of adipose cell numbers in PE and NPE tumors. $n=16$. **b**, Heat map of *ADIPOQ* expression (unique list) compared with Hallmark Adipogenesis, Hallmark Myogenesis, KEGG Vascular smooth muscle contraction and KEGG ECM receptor interaction. Heat map is ranked by *ADIPOQ* values.

21. Line 345: ... We have successfully developed the first, spontaneously occurring model of pleomorphic RMS in mice by lowering the expression levels of the lysosomal sialidase Neu1 in the *Ptch1^{+/-}/ETV7^{TG+/-}* genetic background.....What does “spontaneously” mean? Does it mean without the use of conditional alleles? This is important because there are other models for pleomorphic RMS (e.g. see "Cooperation of oncogenic K-ras and p53 deficiency in pleomorphic rhabdomyosarcoma development in adult mice" published 2006 by Tsumura H and colleagues).

R. Yes, this means without the use of conditional alleles. We have clarified this in the manuscript Line 446-448: “By lowering the expression levels of the lysosomal sialidase Neu1 in the *Ptch1^{+/-}/ETV7^{TG+/-}* genetic background, we have successfully developed the first spontaneously occurring mouse model of pleomorphic RMS without the use of conditional alleles.”

22. Line 374:Furthermore, the overlooked adipose component in these tumors underscores the importance of combining the use of adipose markers during diagnosis of this type of RMS....please discuss these data in light of the paper “A mouse model of rhabdomyosarcoma originating from the adipocyte lineage” published by Mark E. Hatley and colleagues.

R. The paper the reviewer refers to is the original Hatley paper, in which he uses AP2-Cre to drive a lox-stop-lox-Smo construct in adipogenic progenitors. In a more recent paper (Drummond, C.J. *et al.* Hedgehog Pathway Drives Fusion-Negative Rhabdomyosarcoma Initiated from Non-myogenic Endothelial Progenitors. *Cancer Cell*. 2018 Jan 8;33(1):108-124.e5. doi: 10.1016/j.ccell.2017.12.001), the Hatley group describes that the AP2-Cre driver is also expressed in endothelial cells, which they go on to show is the tumor progenitor cell, not the adipose cells. Therefore, our adipose signature cannot be related to the Hatley paper.

23. Line 382: ... Neu1+/- FVB/NJ mice...please provide a reference

R. We have added the reference to the manuscript (Line 494). "*de Geest et al. Systemic and neurologic abnormalities distinguish the lysosomal disorders sialidosis and galactosialidosis in mice. Human Molecular Genetics, 2002 Vol. 11 No 12.*"

Minor:

24. Figure legends: it would be easier for the reader if a)b)c) etc are consistently either in front or at the end of a sentence

R. We have moved the "a,b,c" to the front of the sentence.

25. Fig. 1: please omit **** $P \leq 0.0001$

R. We have removed **** $P \leq 0.0001$

26. Fig. 4: "c)" is missing

R. We have added c,

Reviewer #2 (Remarks to the Author):

Review of Machado et al, "Modelling pleomorphic rhabdomyosarcoma in mice by haploinsufficiency of the lysosomal sialidase NEU1"

Summary

This work builds off of previous work from the corresponding authors, either studying lysosomal exocytosis in a *Arf*^{-/-};*Neu*^{+/-} deficient mouse models of pleomorphic sarcomas, or ETV7 and its role in rapamycin insensitive mTOR complexes in many cancers, including rhabdomyosarcoma (RMS). This collaborative effort implements a new combination of alleles and genetic mouse modeling of pleomorphic rhabdomyosarcoma, a rare RMS sub-type that predominantly occurs in adults, is difficult to treat, and has poor overall survival. This was an exciting new model that had a high incidence of pleomorphic RMS, and generated mechanistic data to detail how NEU1 deficiency skewed developing tumors towards a RMS phenotype. Previous pleomorphic RMS models have relied on oncogenic K-ras and p53 deficiency to generate pleomorphic rhabdomyosarcoma, thus this model introduces new cooperating genetic events that generate this rare tumor.

Specifically, the authors implemented a previous mouse model that had a low incidence of RMS because of *Ptch1* deficiency. They added overexpression of human ETV7 as well as NEU1 deficiency. These allelic combinations are referred to as PE (*Patch*^{+/-}; *ETV7Tg*^{+/-}) or NPE (*Patch*^{+/-}; *ETV7Tg*^{+/-};*Neu*^{+/-}). NEU1 is a lysosomal sialidase that is suppressed in many cancer types, and results in a degradation of plasma membrane integrity and the extracellular matrix. Here, the authors show that *Neu1* deficiency results in an enhanced incidence of pleomorphic RMS, and that NPE tumors have increased fibrosis and collagen staining, and a redistribution of Lamp1 at the plasma membrane of tumor and stromal cells. They perform single cell flow analysis to determine that the tumors are both NP and NPE tumor types are very heterogenous and that Lamp1 plasma membrane positive cells are enriched in exocytic and undifferentiated NPE RMS. Further, they determine that NPE has an adipogenic gene signature associated with less differentiated characteristics that is not present in NP, and suggest that this is a shared feature, especially with the alveolar RMS subtype.

With regards to comparison to the human disease, the authors use a Human 2.0 ST Microarray on their mouse tumor samples, and compare to data from pediatric RMS sub-types, embryonal and alveolar. I recognize that this is because pleomorphic RMS expression data may be unavailable. They show that: 1) NPE vs PE exhibit few gene expression differences, and 2) there are shared features with both ERMS and ARMS. Further, NPE and PE both express the classical clinical markers, desmin, myod, and myogenin, and there is a pleomorphic histology. However, more gene expression analysis should be performed (with pre-existing data) to put this model in the context of other mouse rhabdomyosarcoma models, and enhance the argument for a separate, pleomorphic subtype.

Overall, this is an interesting new mouse model and mechanism to understand the pleomorphic sarcoma sub-type, and to compare how its molecular features are different or shared across other RMS sub-types. This study will be of interest to the field.

Specific Comments:

1) Contextual comparison of pleomorphic RMS to other RMS mouse models. Pleomorphic RMS presents in adult patients, whereas ARMS and ERMS are in pediatric populations. How does the presentation of this model differ from other RMS models? Is the latency similar? Parallels are made with ARMS, and understanding how this model differs from other pediatric sarcoma mouse models could bolster the argument for shared adipogenic features in pleomorphic RMS and ARMS vs. ERMS. To address this, a differential gene expression analysis/PCA/hierarchical clustering of NPE and PE microarray data with previously existing mouse RMS microarray data would support the rationale for this representing pleomorphic RMS/a distinct entity (microarray data from other RMS mouse models is available from Geo Database: GSE22520).

R. The reviewer is referring to the Geo Database GSE22520, that contains microarray data sets of other RMS mouse models. Unfortunately, these data have been generated using an old, non-compatible microarray platform that cannot be compared to our PE and NPE data set.

In addition, the mouse models in this data set were all made with the goal to understand the origin of RMS using several genetic drivers, such as *Pax3*, *Pax7*, *Myf5*, *Myf6*, *PTC1*, and *Rb1* mostly in combination with *Trp53*. Given the function of *Neu1* as a lysosomal sialic acid-cleaving enzyme, our main objective was to investigate the downstream effects of *Neu1* haploinsufficiency caused by posttranslational alterations in the sialic acid content of tumor related proteins. Our results indicate that *Neu1* haploinsufficiency favors the transition of RMS to a more aggressive phenotype rather than inducing tumorigenesis itself.

We have edited the Discussion to reflect this point as follows: (Line 448-456). *“It is important to emphasize that the effect of Neu1^{+/-} is not in the generation of RMS, but rather in its pleomorphic transformation once the tumor is initiated; while ETV7 expression is promoting a high incidence of RMS formation in Ptch1^{+/-} mice³². Although, NPE tumors arise preferentially in the limbs and trunk, and only express 10% of the genes that trended with human ARMS, their genetic features are much closer to ERMS. Unfortunately, the unavailability of compatible genetic data from human pleomorphic RMS that we could compare to our NPE data sets, limits our current evaluation of NPE tumors to their histological characteristics, and further investigation is required to confirm NPE mice as a model of human pleomorphic RMS.”*

2a) Criteria for defining a pleomorphic RMS vs other sub-type diagnosis in the mouse model. Figure 1a details all of the cases for RMS, but it's not clear how these are being defined. I'm assuming this is based off of H&E staining and pathological review, as well as MyoD, MyoG, and Desmin reactivity, but this is not explicitly stated in the text and is an important basis for the paper. Re-write for clarity.

R. Following the Reviewer' suggestion we have revised the text to clearly state the methodology used to diagnose the RMS tumors.

Line 104-107: "RMS was diagnosed based on morphology and immune reactivity to the myogenic regulatory factors (Mrfs) MyoD and Myogenin, and the muscle-specific type III intermediate filament desmin (Des) (Extended Data Fig. 1c-e)."

2b) It would also be interesting to know what the other tumor types were that developed in PE and NPE if this data is available. Were the other tumors types (outside of RMS) similar between NPE and PE, or different?

R. We have observed very few non-RMS tumors, the ones that did occur were not further examined for this study. We found that *Ptch1*^{+/-} mice had 3 solid tumors and 1 brain mass; PE mice had 2 solid tumors and 1 brain mass and NPE mice had 1 solid tumor and 2 brain masses, all of which were not consistent with RMS. In addition, *Ptch1*^{+/-}, PE and NPE mice sporadically developed lymphomas.

It is likely that the brain tumors that grew in the cerebellum were MB, which do occur in *Ptch1*^{+/-} mice (Hahn, H. *et al.* Rhabdomyosarcomas and radiation hypersensitivity in a mouse model of Gorlin syndrome. *Nature medicine* 1998; **4**, 619-622, doi:10.1038/nm0598-619). Given that these were sporadic occurrences and not within the scope of this manuscript, we did not characterize them further.

3) Neu1 deficiency contributes to more aggressive disease in other sarcoma sub-types, but in the PE vs NPE mouse model there is identical survival. Although the NPE tumors have more aggressive features, why do you think this survival is identical? Could the effect of Neu1 be most relevant when mitigating efficacy of cytotoxic chemotherapy agents used for treatment rather than for disease onset. This is outside the scope of this manuscript to address experimentally but is an interesting idea and this discrepancy could be commented on in the discussion.

R. Clearly the most significant contribution of Neu 1 is the promotion of aggressive features not on tumor onset or survival. The reviewer rightly indicates Neu1's effect on exocytosis and chemoresistance as we have reported in an RMS cell line (Machado et al. Regulated lysosomal exocytosis mediates cancer progression. *Science Advances* 2015; Dec 18;1(11):e1500603. Doi: 10.1126), where Neu1 loss of function renders cells resistant to doxorubicin. This point is beyond the scope of this paper, but it is certainly worth testing.

In addition, we have not looked at the development of the tumors over time. Our survival curve includes the ages of the mice when they were euthanized. So, the tumors in the PE mice might progress at slower pace compared to those in the NPE mice, but we did not record that.

Minor Comments:

1) Figure 2C – this is presented as a representative image for an increase in connective tissue in RMS TMAs, but it's not clear what the comparison is to. Also, the composition of RMS tumors on

the TMA is not clear (how many ARMS, ERMS, ssRMS?), or how many of the analyzed tumors had this increase in connective tissue?

R. The reviewer has a valid point. There is no appropriate control on the TMA to compare these sections to. Therefore, we have calculated the percentage of connective tissue per area of each TMA section, ERMS (n= 6), ARMS (n=17) and spindle cell/sclerosing RMS (n=3). We have added the quantification of these results in Extended Data Fig. 2a and have adjusted the text accordingly, line 158-165.

“A similar increase in connective tissue was observed in Masson’s trichrome-stained human tissue microarrays (TMAs) from several RMS patients (ERMS n=16, ARMS n=17, and spindle cell/sclerosing RMS n=3), where 50% of all TMA cores had more than 9% of the total area as collagenous material, with an average of 28.7% in ERMS, 9.7% in ARMS and 23.3% in spindle cell/sclerosing RMS, although the latter only comprised 3 samples (Fig. 2c and Extended Data Fig. 2a). In addition, we observed a substantial number of cells that showed intracytoplasmic collagen as well, with an average of 12.8% in ERMS, 8.5% in ARMS and 13.7% in spindle cell/sclerosing RMS (Extended Data Fig. 2b).”

Extended Data Fig. 2 | Connective tissue expression in normal PE and NPE muscles. a and b, Quantification of ECM/collagen deposition (a) and intracytoplasmic (b) in ARMS, ERMS and spindle cell RMS TMA sections. **c,** Masson’s Trichrome staining of normal PE and NPE muscles. **d and e,** *Col1a2* mRNA expression in normal PE and NPE skeletal muscle. Mean ± s.d.; Welch (unpaired) *t*-test; *n*=12 (PE) and *n*=14 (NPE). **g,** *Col4a1* mRNA expression in normal PE and NPE skeletal muscle. Mean ± s.d.; Welch (unpaired) *t*-test; *n*=11 (PE) and *n*=12 (NPE).

2) Is the Figure 5E-F heatmap for high/low flipped for red/blue? I didn’t see the lines 226-229 being true if not flipped, or it wasn’t clear. Mainly the demarcation from Lamp1 being positive or negative is opposite of what I would think based on the clusters.

R. The reviewer is correct, the heatmap legend has been flipped and has now been corrected.

3) Referencing specific clusters for Figure 5 within the text would help reader understanding

R. We have adjusted the text accordingly to specify the different clusters.

4) Figure 6- the focus of this figure is on overexpressed genes, but were any patterns seen with the NPE downregulated genes?

R. We have now included EnrichR analyses of genes differentially downregulated in NPE vs PE samples in the manuscript (Extended Data Fig. 7c and Supplementary Table 4) as follows:

Line 334-336: "... whereas genes downregulated in NPE belong to the following pathways: matrix metalloproteinases, cytokine-cytokine receptor interaction, HIF-1 signaling pathway and lung fibrosis (Extended Data Fig. 7c , Supplementary Table 4)."

Extended Data Fig. 7 | Analysis of the genetic landscape of NPE versus PE RMS. a, Heat map of murine microarray data comparing overall mRNA expression between PE and NPE RMS samples. A total of 778 genes are shown with $\log_2FC \leq -0.5$ and ≥ 0.5 ; $P \leq 0.5$. L – limb RMS; T

– trunk RMS. **b**, Analysis by Enrichr of genes upregulated in NPE compared with PE RMS. Pathways within the Wiki Pathways 2021 and GO Biological Process 2018 libraries are shown in which NPE upregulated genes were enriched; $P < 0.05$. **c**, Analysis by Enrichr of genes downregulated in NPE compared with PE RMS. Pathways within the Wiki Pathways 2021 and GO KEGG 2018 libraries are shown in which NPE downregulated genes were enriched; $P < 0.05$.

5) In the methods section it says the Human 2.0 ST array was used for mouse tumor samples—what was the rationale? How many of the human microarray probes align to the mouse genome?

R. We have corrected this in the manuscript: “*For microarray analysis of NPE and PE trunk and limb RMS, total RNA (100 ng) was converted into biotin-labeled cRNA (Ambion WT Expression Kit, Affymetrix Inc) and hybridized to Clariom S Mouse GeneChip (Affymetrix Inc) and signals summarized by RMA (Affymetrix Expression Console v1.1).*” (Line 613-615)

6) Line 134 – what myogenic markers are being referred to for determining “differentiation status” in this context.

R. The combination of different observations, such as the histopathological appearance and the protein and RNA expression of MyoD, MyoG and Desmin.

7) Extended data is not referenced in order.

R. The reviewer is correct, we have modified the Extended Data Figure order in the manuscript.

** See the Nature Portfolio author and referees' website at www.nature.com/authors for information about policies, services and author benefits. Communications Biology is committed to improving transparency in authorship. As part of our efforts in this direction, we are now requesting that all authors identified as ‘corresponding author’ create and link their Open Researcher and Contributor Identifier (ORCID) with their account on the Manuscript Tracking System prior to acceptance. ORCID helps the scientific community achieve unambiguous attribution of all scholarly contributions. You can create and link your ORCID from the home page of the Manuscript Tracking System by clicking on ‘Modify my Springer Nature account’ and following the instructions in the link below. Please also inform all co-authors that they can add their ORCIDs to their accounts and that they must do so prior to acceptance. <https://www.springernature.com/gp/researchers/orcid/orcid-for-nature-research>

If you experience problems in linking your ORCID, please contact the Platform Support Helpdesk.

This email has been sent through the Springer Nature Tracking System NY-610A-NPG&MTS *Confidentiality Statement: This e-mail is confidential and subject to copyright. Any unauthorised use or disclosure of its contents is prohibited. If you have received this email in error please notify our Manuscript Tracking System Helpdesk team at <http://platformsupport.nature.com>.*

Details of the confidentiality and pre-publicity policy may be found here <http://www.nature.com/authors/policies/confidentiality.html>

Privacy Policy | Update Profile

REVIEWERS' COMMENTS:

Reviewer #1 (Remarks to the Author):

The authors have answered all my questions and have performed new experiments to support their findings. It is an interesting paper, which provides new insights in the pathology of rhabdomyosarcoma. In my opinion it deserved publication.

Reviewer #2 (Remarks to the Author):

The authors have addressed the reviewer comments satisfactorily and have presented new data that has improved the overall study. I'm happy to recommend publication.